# Score as Action: Fine-Tuning Diffusion Generative Models by Continuous-time Reinforcement Learning

**Hanyang Zhao** [1]  **Haoxian Chen** [1]  **Ji Zhang** [2]  **David D. Yao** [1]  **Wenpin Tang** [1]

## Abstract

Reinforcement learning from human feedback (RLHF), which aligns a diffusion model with input prompt, has become a crucial step in building reliable generative AI models. Most works in this area use a *discrete-time* formulation, which is prone to induced discretization errors, and often not applicable to models with higher-order/black-box solvers. The objective of this study is to develop a disciplined approach to fine-tune diffusion models using *continuous-time* RL, formulated as a stochastic control problem with a reward function that aligns the end result (terminal state) with input prompt. The key idea is to treat score matching as controls or actions, and thereby making connections to policy optimization and regularization in continuous-time RL. To carry out this idea, we lay out a new policy optimization framework for continuous-time RL, and illustrate its potential in enhancing the value networks design space via leveraging the structural property of diffusion models. We validate the advantages of our method by experiments in downstream tasks of fine-tuning large-scale Text2Image models of Stable Diffusion v1.5.

## 1. Introduction

Diffusion models (Sohl-Dickstein et al., 2015), with the capacity to turn a noisy/non-informative initial distribution into a desired target distribution through a well-designed denoising process (Ho et al., 2020; Song et al., 2020; 2021b), have recently found applications in diverse areas such as high-quality and creative image generation (Ramesh et al., 2022; Shi et al., 2020; Saharia et al., 2022; Rombach et al., 2022), video synthesis (Ho et al., 2022), and drug design

(Xu et al., 2022). And, the emergence of human-interactive platforms like ChatGPT (Ouyang et al., 2022) and Stable Diffusion (Rombach et al., 2022) has further increased the demand for diffusion models to align with human preference or feedback.

To meet such demands, (Hao et al., 2022) proposed a natural way to fine-tune diffusion models using reinforcement learning (RL, (Sutton & Barto, 2018)). Indeed, RL has already demonstrated empirical successes in enhancing the performance of LLM (large language models) using human feedback (Christiano et al., 2017; Ouyang et al., 2022; Bubeck et al., 2023), and (Fan & Lee, 2023) is among the first to utilize RL-like methods to train diffusion models for better image synthesis. Moreover, (Lee et al., 2023; Fan et al., 2023; Black et al., 2023) have improved the text-to-image (T2I) diffusion model performance by incorporating reward models to align with human preference (e.g., CLIP (Radford et al., 2021), BLIP (Li et al., 2022), ImageReward (Xu et al., 2024)).

Notably, all studies referenced above that combine diffusion models with RL are formulated as *discrete-time* sequential optimization problems, such as Markov decision processes (MDPs, (Puterman, 2014)), and solved by discrete-time RL algorithms such as REINFORCE (Sutton et al., 1999) or PPO (Schulman et al., 2017).

Yet, diffusion models are intrinsically *continuous-time* as they were originally created to model the evolution of thermodynamics (Sohl-Dickstein et al., 2015). Notably, the continuous-time formalism of diffusion models provides a unified framework for various existing discrete-time algorithms as shown in (Song et al., 2021b): the denoising steps in DDPM (Ho et al., 2020) can be viewed as a discrete approximation of a stochastic differential equation (SDE) and are implicitly *score-based* under a specific variance-preserving SDE (Song et al., 2021b); and DDIM (Song et al., 2020), which underlies the success of Stable Diffusion (Rombach et al., 2022), can also be seen as a numerical integrator of an ODE (ordinary differential equation) sampler (Salimans & Ho, 2022). Awareness of the continuous-time nature informs the design structure of the discrete-time SOTA large-scale T2I generative models (e.g.,(Dhariwal & Nichol, 2021; Rombach et al., 2022; Esser et al., 2024)), and

---

[1]Department of IEOR, Columbia University, New York, USA. [2]Department of CS, Stony Brook University, New York, USA. Correspondence to: <hz2684, yao, wt2319@columbia.edu>.

*Proceedings of the 42$^{st}$ International Conference on Machine Learning*, Vancouver, Canada. PMLR 267, 2025. Copyright 2025 by the author(s).

enables simple controllable generations by classifier guidance to solve inverse problems (Song et al., 2021b;a). It also motivates more efficient diffusion models with continuous-time samplers, including the ODE-governed probability (normalizing) flows (Papamakarios et al., 2021; Song et al., 2021b) and rectified flows (Liu et al., 2022; 2023) underpinning Stable Diffusion v3 (Esser et al., 2024). A discrete-time formulation of RL algorithms for fine-tuning diffusion models, if/when directly applied to continuous-time diffusion models via discretization, can *nullify* the models' continuous nature and fail to capture or utilize their structural properties.

For fine-tuning diffusion models, discrete-time RL algorithms (such as DDPO) require a prior chosen time discretization in sampling. We thus examine the robustness of a fine-tuned model to the inference time discretization, and observe an "overfitting" phenomenon as illustrated in Figure 1. Specifically, improvements observed during inference at alternative discretization timesteps (25 and 100) are significantly smaller than that of sampling timestep (50) in RL.

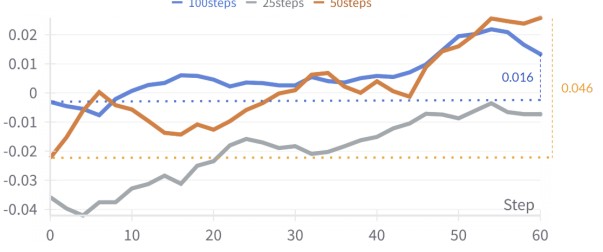

*Figure 1.* Reward curve of model checkpoints sampling under different discretization steps (25, 50, 100): After training Stable Diffusion v1.4 for a fixed prompt with 60 training steps by DDPO (Black et al., 2023) with 50 discretization steps, the average reward of images generated by the checkpoints obtained (under 50 discretization steps) evaluated by ImageReward (Xu et al., 2024) increases by 0.046, while the average reward of images generated with 100 discretization steps only increases by less than 0.016.

In addition, for high-order solvers (such as 2nd order Heun used in EDM (Karras et al., 2022)), discrete-time RL methods will require solving a high-dimension root-finding problem for each inference step, which is inefficient in practice.

**Main contributions.** To address the above issues, we develop a unified continuous-time RL framework to fine-tune score-based diffusion models.

Our first contribution is a continuous-time RL framework for fine-tuning diffusion models by treating *score functions as actions*. This framework naturally accommodates discrete-time diffusion models with any solver as well as continuous-time diffusion models, and overcomes the afore-mentioned limitations of discrete-time RL methods. (See Section 3.)

Second, we illustrate the promise of leveraging the structural property of diffusion models to generate tractable optimization problems and to enhance the design space of value networks. This includes transforming the KL regularization to a tractable running reward over time, and a novel design of value networks that involves "sample prediction" (also known as $x$-prediction in diffusion literature) by sharing parameters with policy networks and fine-tuned diffusion models. Through experiments, we demonstrate the drastic improvements over naive value network designs.

Third, we provide a new theory for RL in continuous-time and space, which leads to the first scalable policy optimization algorithm (for continuous-time RL), along with estimates of policy gradients via generated samples. Compared with existing works in continuous-time PPO, we consider a special case of state-independent diffusion coefficients cater for the diffusion models design, which yields sharper bounds and closed-form advantage-rate functions instead of estimations. (See Section 4.)

### 1.1. Related Works

Papers that relate to our work are briefly reviewed below.

**Continuous-time RL**. (Wang et al., 2020) models the noise or randomness in the environment dynamics as following an SDE, and incorporates an entropy-based regularizer into the objective function to facilitate the exploration-exploitation tradeoff. Follow-up works include designing model-free methods and algorithms under either finite horizon (Jia & Zhou, 2022a;b; 2023) or infinite horizon (Zhao et al., 2024b).

**RL for fine-tuning T2I diffusion models**. DDPO (Black et al., 2023) and DPOK (Fan et al., 2023) both discrete the time steps and fine-tune large pretrained T2I diffusion models through the reinforcement learning algorithms. Moreover, (Ren et al., 2024) introduces DPPO, a policy gradient-based RL framework for fine-tuning diffusion-based policies in continuous control and robotic tasks.

**Other Preference Optimizations for diffusion models**. (Wallace et al., 2024) proposes an adaptation of Direct Preference Optimization (DPO) for aligning T2I diffusion models like Stable Diffusion XL to human preferences. (Yuan et al., 2024) proposes a novel fine-tuning method for diffusion models that iteratively improves model performance through self-play, where a model competes with its previous versions to enhance human preference alignment and visual appeal. See Section 4.5 in (Winata et al., 2024) for a review.

**Stochastic Control**. (Uehara et al., 2024), which also formulated the diffusion models alignment as a continuous-time stochastic control problem with a different parameterization of the control; (Tang, 2024) also provides a more rigorous review and discussion. (Domingo-Enrich et al., 2024) proposes to use adjoint methods instead to solve the

similar control problem. In a concurrent work to ours, (Gao et al., 2024) uses $q$-learning (Jia & Zhou, 2023) for inferring the score of diffusion models (instead of fine tuning a pretrained model), whose formulation relies on an earlier version (Zhao et al., 2024a) of this paper.

The rest of the paper is organized as follows. In Section 2, we review the preliminaries of continuous-time RL and score-based diffusion models. Section 3 presents our continuous-time framework for fine-tuning diffusion models using RLHF, with the theory and algorithm for policy optimization detailed in Section 4, and the effectiveness of the algorithm illustrated in Section 5. Concluding remarks and discussions are presented in Section 6.

## 2. Preliminaries

### 2.1. Continuous-time RL

**Diffusion Process.** We consider the state space $\mathbb{R}^d$, and denote by $\mathcal{A}$ the action space. Let $\pi(\cdot \mid t, x)$ be a feedback policy given $t \in [0, T]$ and $x \in \mathbb{R}^d$. The state dynamics $(X_t^\pi, 0 \leq t \leq T)$ is governed by the following SDE:

$$\mathrm{d}X_t^\pi = b(t, X_t^\pi, a_t)\,\mathrm{d}t + \sigma(t)\mathrm{d}B_t, \quad X_0^\pi \sim \rho, \quad (1)$$

where $(B_t, t \geq 0)$ is a $d$-dimensional Brownian motion; $b : \mathbb{R}_+ \times \mathbb{R}^d \times \mathcal{A} \to \mathbb{R}^d$ and $\sigma : \mathbb{R}_+ \to \mathbb{R}_+$ [1] are given functions; the action $a_t$ follows the distribution $\pi(\cdot \mid t, X_t^\pi)$ by external randomization; and $\rho$ is the initial distribution over the state space.

**Performance Metric.** Our goal is to find the optimal feedback policy $\pi^*$ that maximizes the expected reward over a finite time horizon:

$$V^* := \max_\pi \mathbb{E}\left[\int_0^T r(t, X_t^\pi, a_t^\pi)\,\mathrm{d}t + h(X_T^\pi) \mid X_0^\pi \sim \rho\right], \quad (2)$$

where $r : \mathbb{R}_+ \times \mathbb{R}^d \times \mathcal{A} \to \mathbb{R}$ and $h : \mathbb{R}^d \to \mathbb{R}$ are the running and terminal rewards respectively. Given a policy $\pi(\cdot)$, let $\tilde{b}(t, x, \pi(\cdot)) := \int_\mathcal{A} b(t, x, a)\pi(a)\mathrm{d}a$. We consider the following equivalent representation of (1):

$$\mathrm{d}\tilde{X}_t = \tilde{b}\left(t, \tilde{X}_t, \pi(\cdot \mid t, \tilde{X}_t)\right)\mathrm{d}t + \sigma(t)\mathrm{d}\tilde{B}_t, \quad \tilde{X}_0 \sim \rho, \quad (3)$$

in the sense that there exists a probability measure $\tilde{\mathbb{P}}$ that supports a $d$-dimensional Brownian motion $(\tilde{B}_t, t \geq 0)$, and for each $t \geq 0$, the distribution of $\tilde{X}_t$ under $\tilde{\mathbb{P}}$ agrees with that of $X_t$ under $\mathbb{P}$ defined by (1). Note that the dynamics (3) does not require external randomization. Accordingly, set $\tilde{r}(t, x, \pi) := \int_\mathcal{A} r(t, x, a)\pi(a)\mathrm{d}a$.

---

[1] For our applications here we assume that the diffusion coefficient $\sigma(t)$ only depends on time $t$. Note, however, that the general continuous-time RL theory also holds for time-, state- and action-dependent $\sigma(t, x, a)$, see (Jia & Zhou, 2022a;b).

The value function associated with the feedback policy $\{\pi(\cdot \mid t, x) : x \in \mathbb{R}^d\}$ is

$$V(t, x; \pi) := \mathbb{E}\left[\int_t^T r(s, X_s^\pi, a_s^\pi)\,\mathrm{d}s + h(X_T^\pi) \mid X_t^\pi = x\right]$$

$$= \mathbb{E}\left[\int_t^T \tilde{r}(s, \tilde{X}_s^\pi, \pi(\cdot|s, \tilde{X}_s^\pi))\mathrm{d}s + h(\tilde{X}_T^\pi) \mid \tilde{X}_t^\pi = x\right] \quad (4)$$

The performance metric is $V^\pi := \int_{\mathbb{R}^d} V(0, x; \pi)\rho(dx)$, and $V^* := \max_\pi V^\pi$. The task is to construct a sequence of (feedback) policies $\pi_k$, $k = 1, 2, \ldots$ recursively such that the performance metric is non-decreasing in $k$.

$q$**-Value.** Following the definition in (Jia & Zhou, 2023), given a policy $\pi$ and $(t, x, a) \in [0, \infty) \times \mathbb{R}^n \times \mathcal{A}$, we construct a "perturbed" policy, denoted by $\hat{\pi}$: It takes the action $a \in \mathcal{A}$ on $[t, t + \Delta t)$, and then follows $\pi$ on $[t + \Delta t, \infty)$. Specifically, the corresponding state process $X^{\hat{\pi}}$, given $X_t^{\hat{\pi}} = x$, breaks into two pieces: on $[t, t + \Delta t)$, it is $X^a$ following (1) with $a_t \equiv a$ (i.e., $\pi(t, x, a) = 1$); while on $[t + \Delta t, \infty)$, it is $X^\pi$ following (3) but with the initial time-state pair $(t + \Delta t, X_{t+\Delta t}^a)$. The $q$-value measures the rate of the performance difference between the two policies when $\Delta t \to 0$, and is shown in (Jia & Zhou, 2023) to take the following form:

$$q(t, x, a; \pi) = \frac{\partial V}{\partial t}(t, x; \pi) +$$
$$\mathcal{H}\left(t, x, a, \frac{\partial V}{\partial x}(t, x; \pi), \frac{\partial^2 V}{\partial x^2}(t, x; \pi)\right), \quad (5)$$

where $\mathcal{H}(t, x, a, y, A) := b(t, x, a) \cdot y + \frac{1}{2}\sigma^2(t)\sum_i A_{ii} + r(t, x, a)$ is the (generalized) Hamilton function in stochastic control theory (Yong & Zhou, 1999).

### 2.2. Score-Based Diffusion Models

**Forward and Backward SDE.** We follow the presentation in (Tang & Zhao, 2024). Consider the following SDE that governs the dynamics of a process $(X_t, 0 \leq t \leq T)$ in $\mathbb{R}^d$ (Song et al., 2021b),

$$\mathrm{d}X_t = f(t, X_t)\mathrm{d}t + g(t)\mathrm{d}B_t, \quad X_0 \sim p_{\text{data}}(\cdot), \quad (6)$$

where $(B_t, t \geq 0)$ is a $d$-dimensional Brownian motion, $f : \mathbb{R}_+ \times \mathbb{R}^d \to \mathbb{R}^d$ and $g : \mathbb{R}_+ \to \mathbb{R}_+$ are two given functions (up to the designer to choose), and the initial state $X_0$ follows a distribution with density $p_{\text{data}}(\cdot)$, which is shaped by data yet unknown *a priori*. Denote by $p_t(\cdot)$ the probability density of $X_t$.

Run the SDE in (6) until a given time $T > 0$, to obtain $X_T \sim p(T, \cdot)$. Next, consider the "time reversal" of $X_t$, denoted $X_t^{\text{rev}}$, such that the distribution of $X_t^{\text{rev}}$ agrees with

that of $X_{T-t}$ on $[0, T]$. Then, $(X_t^{\text{rev}}, 0 \leq t \leq T)$ satisfies the following SDE under mild conditions on $f$ and $g$:

$$\mathrm{d}X_t^{\text{rev}} = \left(-f(T-t, X_t^{\text{rev}}) + g^2(T-t)\nabla \log p_{T-t}(X_t^{\text{rev}})\right) \\ \mathrm{d}t + g(T-t)\mathrm{d}B_t, \tag{7}$$

where $\nabla \log p_t(x)$ is known as *Stein's score function*. Below we will refer to the two SDE's in (6) and (7), respectively, as the forward and the backward SDE.

For sampling from the backward SDE, we replace $p_T(\cdot)$ with some $p_{\text{noise}}(\cdot)$ as an approximation. The initialization $p_{\text{noise}}(\cdot)$ is commonly independent of $p_{\text{data}}(\cdot)$, which is the reason why diffusion models are known for generating data from "noise".

**Score Matching**. Since the score function $\nabla_x \log p_t(x)$ in (7) is unknown, the idea is to learn the score $s_{\theta_{\text{pre}}}(t, x) \approx \nabla_x \log p_t(x)$, which is often referred to as *pretraining*. It boils down to solving the following denoising score matching (DSM) problem (Vincent, 2011) [2]:

$$\mathcal{J}_{\text{DSM}}(\theta) = \mathbb{E}\left[\lambda(t) \|s_\theta(t, x_t) - \nabla \log p_t(x_t|x_0)\|_2^2\right], \tag{8}$$

where $x_t \sim p_t(\cdot|x_0)$ and $\lambda : [0, T] \to \mathbb{R}_{>0}$ is a chosen positive-valued weight function.

**Inference Process**. Once the best approximation $s_{\theta_{\text{pre}}}$ is obtained, we use it to replace $\nabla \log p_t(x)$ in (7). The corresponding approximation to the reversed process $X_t^{\text{rev}}$, denoted as $X_t^\leftarrow$, then follows the SDE:

$$\mathrm{d}X_t^\leftarrow = \left(-f(T-t, X_t^\leftarrow) + g^2(T-t)s_{\theta_{\text{pre}}}(T-t, X_t^\leftarrow)\right)\mathrm{d}t \\ + g(T-t)\mathrm{d}B_t, \tag{9}$$

with $X_0^\leftarrow \sim p_{\text{noise}}(\cdot)$. At time $t = T$, the distribution of $X_T^\leftarrow$ is expected to be close to $p_{\text{data}}(\cdot)$. The well-known DDPM (Ho et al., 2020) can be viewed as a discretized version of the SDE in (9). This has been established in (Song et al., 2021b; Salimans & Ho, 2022; Zhang & Chen, 2022; Zhang et al., 2022); also refer to further discussions in Appendix A. Throughout the rest of the paper, we will focus on the continuous formalism (via SDE).

## 3. Continuous-time RL for Diffusion Models Fine Tuning

Here we formulate the task of fine-tuning diffusion models as a continuous-time stochastic control problem. The high-level idea it to treat the score function approximation as a control process applied to the backward SDE.

---

[2]There are several existing score matching methods, among which the DSM is the most tractable one because $p_t(\cdot|x_0)$ is accessible for a wide class of diffusion processes; in particular, it is conditionally Gaussian if (6) is a linear SDE (Song et al., 2021b).

**Scores as Actions**. First, to broaden the application context of the diffusion model, we add a parameter $c$ to the score function, interpreted as a "class" index or label (e.g., for input prompts). Then, the backward SDE in (9) becomes:

$$\mathrm{d}X_t^\leftarrow = \left(-f(T-t, X_t^\leftarrow) + g^2(T-t)s_{\theta_{\text{pre}}}(T-t, X_t^\leftarrow, c)\right) \\ \mathrm{d}t + g(T-t)\mathrm{d}B_t. \tag{10}$$

Next, comparing the continuous RL process in (3) and the inference process (10), we choose $b$ and $\sigma$ in the RL dynamics in (3) as:

$$\begin{cases} \sigma(t) := g(T-t), \\ b(t, x, a) := -f(T-t, x) + g^2(T-t)a. \end{cases} \tag{11}$$

In the sequel, we will stick to this definition of $b$ and $\sigma$.

Define a specific feedback control, $a_t^{\theta_{\text{pre}}} = s_{\theta_{\text{pre}}}(T-t, X_t^\leftarrow, c)$, and the backward SDE in (10) is expressed as:

$$\mathrm{d}X_t^\leftarrow = b\left(t, X_t^\leftarrow, a_t^{\theta_{\text{pre}}}\right)\mathrm{d}t + \sigma(t)\mathrm{d}B_t. \tag{12}$$

This way, the score function is replaced by the action (or control/policy), and finding the optimal score becomes a policy optimization problem in RL. Denote by $p^{\theta_{\text{pre}}}(t, \cdot, c)$ the probability density of $X_t^\leftarrow$ in (12).

**Exploratory SDEs**. As we will deal with the time-reversed process $X_t^\leftarrow$ exclusively from now on, the superscript $\leftarrow$ will be dropped to lighten the notation. To enhance exploration, we will use a Gaussian control:

$$a_t^\theta \sim \pi^\theta(\cdot \mid t, X_t^\theta, c) = N(\mu^\theta(t, X_t^\theta, c), \Sigma_t). \tag{13}$$

Specifically, the dependence on $\theta$ is through that of the mean function $\mu^\theta$, while the covariance matrix $\Sigma_t$ only depends on time $t$, representing a chosen exploration level at $t$. For brevity, write $X_t^\theta$ for the (time-reversed) process $X_t^{\pi^\theta}$ driven by the policy $\pi^\theta$. Then $(X_t^\theta, 0 \leq t \leq T)$ is governed by the SDE:

$$\mathrm{d}X_t^\theta = \left[-f(T-t, X_t^\theta) + g^2(T-t)\mu^\theta(t, X_t^\theta, c)\right]\mathrm{d}t \\ + g(T-t)\mathrm{d}B_t, \quad X_0^\theta \sim \rho. \tag{14}$$

Denote by $p^\theta(t, \cdot, c)$ the probability density of $X_t^\theta$.

**Objective Function**. The objective function of the RL problem consists of two parts. The first part is the terminal reward, i.e., a given reward model (RM) that is a function of both $X_T$ and $c$. For instance, if the task is T2I generation, then $\text{RM}(X_T, c)$ represents how well the generated image $X_T$ aligns with the input prompt $c$. The second part is a penalty (i.e., regularization) term, which takes the form of the KL divergence between $p^\theta(T, \cdot, c)$ and its pretrained counterpart. This is similar in spirit to previous works on fine-tuning diffusion models by discrete-time RL, see e.g.,

(Ouyang et al., 2022; Fan et al., 2023). As for exploration, note that it has been represented by the Gaussian noise in $a_t^\theta$; refer to (13), and more on this below. So, here is the problem we want to solve:

$$\max_\theta \mathbb{E}\left[\mathrm{RM}(c, X_T^\theta) - \beta\,\mathrm{KL}\left(p^\theta(T, \cdot, c)\|p^{\theta_{pre}}(T, \cdot, c)\right)\right],$$
(15)

where $\beta > 0$ is a (given) penalty cost.

To connect the problem in (15) to the objective function of the RL model in (2), we need the following explicit expression for the KL divergence term in (15).

**Theorem 3.1.** *For any given c, the KL divergence between $p^\theta$ and $p^{\theta_{pre}}$ is:*

$$\mathrm{KL}(p^\theta(T, \cdot, c)\|p^{\theta_{pre}}(T, \cdot, c))$$
$$= \mathbb{E}\int_0^T \frac{g^2(T-t)}{2}\|\mu^\theta(t, X_t^\theta, c) - \mu^{\theta_{pre}}(t, X_t^\theta, c)\|^2 dt.$$
(16)

*Proof Sketch.* The full proof is given in Appendix B.1.

As a remark, it is important to use the "reverse"-KL divergence $\mathrm{KL}\left(p^\theta(T, \cdot, c)\|p^{\theta_{pre}}(T, \cdot, c)\right)$, because it yields the expectation under the current policy $\pi^\theta$ that can be estimated from sample trajectories. By Theorem 3.1, the objective function in (15) is equivalent to the following:

$$\eta^\theta := \mathbb{E}\int_0^T \underbrace{-\frac{\beta}{2}g^2(T-t)\|\mu_t^\theta - \mu_t^{\theta_{pre}}\|^2}_{r(t, X_t^\theta, a_t^\theta)} dt$$
$$+ \mathbb{E}\underbrace{\mathrm{RM}(X_T^\theta, c)}_{h(X_T^\theta, c)},$$
(17)

where we abbreviate $\mu^\theta(t, X_t^\theta, c)$ and $\mu^{\theta_{pre}}(t, X_t^\theta, c)$ by $\mu_t^\theta$ and $\mu_t^{\theta_{pre}}$ respectively. Thus, maximizing the objective function in (15) aligns with the RL model formulated in (2). We can also define the corresponding value function as:

$$V^\theta(t, x; c) = \mathbb{E}\left[\int_t^T -\frac{\beta}{2}g^2(T-t)\|\mu_t^\theta - \mu_t^{\theta_{pre}}\|^2 dt\right.$$
$$\left. + \mathrm{RM}(X_T^\theta, c) \mid X_t^\theta = x\right],$$
(18)

**Value Network Design**. We also adopt a function approximation to learn the value function (i.e., the critic). For the value function $V^\theta(t, x; c)$ associated with policy $\pi^\theta$, there is the boundary condition:

$$V^\theta(T, x; c) = \mathbb{E}\left[\mathrm{RM}(X_T^\theta, c) \mid X_T^\theta = x\right] = \mathrm{RM}(x, c).$$
(19)

To meet this condition, we propose the following parametrization that leverages the structural property of

diffusion models:

$$V^\theta(t, x; c) \approx \mathcal{V}_\phi^\theta(t, x; c) :=$$
$$\underbrace{c_{\mathrm{skip}}(t) \cdot \mathrm{RM}(\hat{x}_\theta(t, x, c))}_{\text{reward mean predictor}} + \underbrace{c_{\mathrm{out}}(t) \cdot F_\phi(t, x, c)}_{\text{residual term corrector}},$$
(20)

where $\mathcal{V}_\phi^\theta$ denotes a family of functions parameterized by $(\theta, \phi)$, and

$$\hat{x}_\theta(t, x, c) = \frac{1}{\alpha_t}\left(\sigma_t^2 s_\theta(t, x, c) + x\right),$$
(21)

with $\alpha_t$ and $\sigma_t$ being noise schedules of diffusion models (see Appendix A.2 for details). When $\theta = \theta_{\mathrm{pre}}$, $\hat{x}_\theta$ predicts a denoised sample given the current $x$ and the score estimate $s_\theta(t, x, c)$, which is known as *Tweedie's formula*. To treat the second term in (18), our intuition comes from that

$$\mathrm{RM}(\mathbb{E}(X_T \mid X_t)) \approx \mathbb{E}(\mathrm{RM}(X_T) \mid X_t),$$
(22)

if we are allowed to exchange the conditional expectation and the reward model score (though generally it's not true). $F_\phi(t, x, c)$ are effectively approximations to the residual term, which can be seen as a composition of the possible reward error and the first term in (18).

We refer these two parts to as *reward mean predictor* and *residual corrector*. There $c_{\mathrm{skip}}(t)$ and $c_{\mathrm{out}}(t)$ are differentiable functions such that $c_{\mathrm{skip}}(T) = 1$ and $c_{\mathrm{out}}(T) = 0$, so the boundary condition (19) is satisfied. Notably, similar parametrization trick has also been used to train successful diffusion models such as EDM (Karras et al., 2022) and consistency models (Song et al., 2023).

For learning the value function, we use trajectory-wise Monte Carlo estimation to update $\phi$ by minimizing the mean square error (MSE). In our experiments, we observe that choosing $c_{\mathrm{skip}}(t) = \cos(\frac{\pi}{2T}t)$ and $c_{\mathrm{out}}(t) = \sin(\frac{\pi}{2T}t)$ yields the smallest loss (see Table 1). Also refer to Section 5.2 for more architecture details.

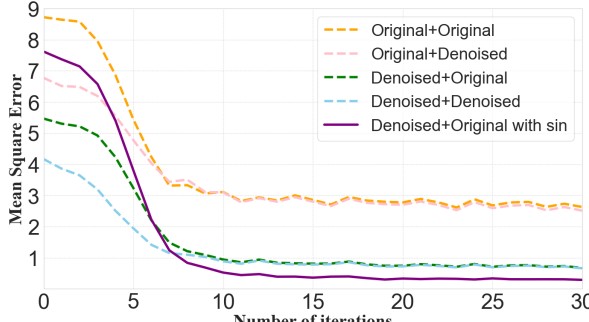

*Figure 2.* Pretraining Value Function with Different Architecture.

Note that it is possible to learn the value function by either solving the associated Hamilton-Jacobi-Bellman equation,

| Architecture | Predictor | Corrector | $c_{\text{out}}$ | MSE |
|---|---|---|---|---|
| Baseline | $RM(x, c)$ | $F(x, c)$ | $1\text{-}\cos(\frac{\pi}{2T}t)$ | 2.63 |
| Org+Denoised | $RM(x, c)$ | $F(\hat{x}_\theta, c)$ | $1\text{-}\cos(\frac{\pi}{2T}t)$ | 2.51 |
| Denoised+Orig | $RM(\hat{x}_\theta, c)$ | $F(x, c)$ | $1\text{-}\cos(\frac{\pi}{2T}t)$ | 0.67 |
| Denoised+Denoised | $RM(\hat{x}_\theta, c)$ | $F(\hat{x}_\theta, c)$ | $1\text{-}\cos(\frac{\pi}{2T}t)$ | 0.66 |
| **Denoised+Orig** | $RM(\hat{x}_\theta, c)$ | $F(x, c)$ | $\sin(\frac{\pi}{2T}t)$ | **0.29** |

*Table 1.* Comparison of different architecture configurations: $\hat{x}_\theta$ is abbreviated for $\hat{x}_\theta(t, x, c)$.

or minimizing Bellman's error rate (which can be seen as the continuous-time analog of temporal difference). However, both approaches will yield a supervised learning objective that contains the second-order derivative of the value function, which is hard to optimize. We leave for future work the investigation of policy evaluation methods that are based on partial differential equations or temporal difference rates.

## 4. Continuous-time Policy Optimization

To efficiently optimize the continuous-time RL problem raised above, we further develop the theory of policy optimization in continuous time and space for fine-tuning diffusion models. Different from the general formalism in the literature (Schulman et al., 2015; Zhao et al., 2024b), we focus on the case of (1) KL regularized rewards, and (2) state-independent diffusion coefficients in the continuous-time setup, which yield new results not only in the analysis but also in the resulting algorithms.

**Policy Gradient**. We first show that the continuous-time policy gradient can be directly computed without any prior discretization of the time variable.

**Theorem 4.1.** *The gradient of an admissible policy $\pi^\theta$ parameterized by $\theta$ takes the form:*

$$\nabla_\theta V^\theta = \mathbb{E}\left[\int_0^T \nabla_\theta \log \pi^\theta(a_t^\theta|t, X_t^\theta) q(t, X_t^\theta, a_t^\theta; \pi^\theta)\mathrm{d}t\right],$$
(23)

*where $\pi^\theta$, $a_t^\theta$ and $q$ are as defined in* (13) *and* (5).

*Proof Sketch.* The full proof is given in Appendix B.2.

Note that the only terms in the $q$-value function that involve action $a$ are (the second order term is irrelevant to action $a$):

$$g^2(T - t)a\frac{\partial V^\theta}{\partial x}(t, x) =: \tilde{q}^\theta(t, x, a).$$

In addition, the value function approximation can be computed by Monte Carlo or the martingale approach as in Jia & Zhou (2022a), and then $\frac{\partial V}{\partial x}$ can be evaluated by backward propagation. Since the reward can be non-differentiable, and also for the sake of efficient computation, we can approximate $\tilde{q}^\theta(t, x, a) \approx \left(V(t, x + \eta\, g^2(T - t)a) - V(t, x)\right)/\eta$, where $\eta$ is a scaling parameter.

**Continuous-time TRPO/PPO**. We also derive the continuous-time finite horizon analogies of TRPO and PPO for the discrete RL in the finite horizon setting (Schulman et al., 2015; 2017), and in the continuous-time infinite horizon setup (Zhao et al., 2024b). The Performance Difference Lemma (PDL) is as follows.

**Lemma 4.2.** *We have that:*

$$V^{\hat{\theta}} - V^\theta = \mathbb{E}\int_0^T q(t, X_t^{\hat{\theta}}, a_t^{\hat{\theta}}; \pi^\theta)\mathrm{d}t.$$
(24)

The proof is similar to Theorem 2 in (Zhao et al., 2024b). In the same essence of (Kakade & Langford, 2002; Schulman et al., 2015; Zhao et al., 2024b), we define the *local approximation function* to $V^{\hat{\theta}}$ by

$$L^\theta(\hat{\theta}) = V^\theta + \mathbb{E}\int_0^T \frac{\pi^{\hat{\theta}}(a_t^\theta|t, X_t^\theta)}{\pi^\theta(a_t^\theta|t, X_t^\theta)} q(t, X_t^\theta, a_t^\theta; \pi^\theta)\mathrm{d}t.$$
(25)

Observe that

(i) $L^\theta(\theta) = V^\theta,$ (ii) $\nabla_{\hat{\theta}} L^\theta(\hat{\theta})\mid_{\hat{\theta}=\theta} = \nabla_{\hat{\theta}} V^{\hat{\theta}}\mid_{\hat{\theta}=\theta},$

i.e., the local approximation function and the true performance objective share the same value and the same gradient with respect to the policy parameters. Thus, the local approximation function can be regarded as the first order approximation to the performance metric.

Now we provide analysis on the gap $V^{\hat{\theta}} - L^\theta(\hat{\theta})$, which guarantees the policy improvement (similar to approaches for discounted/average reward MDP (Schulman et al., 2015; Zhang & Ross, 2021), and for continuous-time RL in the infinite horizon (Zhao et al., 2024b)).

**Assumption 4.3.** (*Bounded Reward and q function*): There exists $M > 0$ such that for any $c$, $x$ and $a$, $|RM(x, c)| \leq M$ and $|q(t, x, a; \pi^\theta)| \leq M$ for any $t$, $\pi^\theta$.

**Theorem 4.4.** *Under Assumption 4.3, then for any policy $\hat{\theta}$ such that $\left|\ln\left(\frac{p^{\hat{\theta}}(x)}{p^{\theta_{\text{pre}}}(x)}\right)\right|$ is bounded, and $\text{KL}(p^\theta \parallel p^{\hat{\theta}}) \leq 1$, there exists a constant $C > 0$ such that:*

$$|V^{\hat{\theta}} - L^\theta(\hat{\theta})| \leq$$

$$C\left(\mathbb{E}\int_0^T \text{KL}(\pi^\theta(\cdot|t, X_t^\theta)\|\pi^{\hat{\theta}}(\cdot|t, X_t^\theta))\mathrm{d}t\right)^{\frac{1}{2}}.$$
(26)

*Proof Sketch.* Different from the proofs in (Schulman et al., 2015; Zhao et al., 2024b), which bound the difference through PDL, our proof relies on the structural property of diffusion models by bounding:

$$|V^{\hat{\theta}} - V^\theta| \text{ and } |L^\theta(\hat{\theta}) - V^\theta|$$

separately. The detailed proof is given in Appendix B.3.

By Theorem 4.4, we can apply the same technique as in PPO (Schulman et al., 2017) by clipping the ratio and replacing $q$

with $\tilde{q}$ (which is equivalent to adapting a baseline function). This yields the policy update rule as:

$$\theta_{n+1} = \max_{\theta} \mathbb{E} \int_0^T \min\left(\rho_t^\theta q_t^{\theta_n}, \mathrm{clip}\left(\rho_t^\theta, \epsilon\right) q_t^{\theta_n}\right) \mathrm{d}t, \quad (27)$$

where the advantage rate function and the likelihood ratio are defined by

$$q_t^{\theta_n} = \tilde{q}(t, X_t^{\theta_n}, a_t^{\theta_n}; \pi_n^\theta), \quad \rho_t^\theta = \frac{\pi^\theta(a_t^{\theta_n}|t, X_t^{\theta_n})}{\pi^{\theta_n}(a_t^{\theta_n}|t, X_t^{\theta_n})}.$$

The surrogate objective can then be optimized by stochastic gradient descent. The pseudo-code of our algorithm is listed in Appendix C.2.

## 5. Experiments

### 5.1. Enhancing Small-Steps Diffusion Models

**Setup**. We evaluate the ability of our proposed algorithm to train short-run diffusion models with significantly reduced generation steps $T$, while maintaining high sample quality. In the experiment, we take $T = 10$. Our experiments are conducted on the CIFAR-10 (32×32) dataset (Krizhevsky et al., 2009). We fine-tune pretrained diffusion model backbone using DDPM (Ho et al., 2020). The primary evaluation metric is the Fréchet Inception Distance (FID) (Heusel et al., 2017), which measures the quality of generated samples.

To benchmark our method, we compare it against DxMI (Yoon et al., 2024), which formulates the diffusion model training as an inverse reinforcement learning (IRL) problem. DxMI jointly trains a diffusion model and an energy-based model (EBM), where the EBM estimates the log data density and provides a reward signal to guide the diffusion process. To ensure a fair comparison, we replace the policy improvement step in DxMI with our continuous-time RL counterpart, maintaining consistency while evaluating the effectiveness of our approach. We set the learning rate of the value network to $2 \times 10^{-5}$ and U-net to $3 \times 10^{-7}$.

**Result**. Figure 3 shows our approach converges significantly faster than DxMI, and achieves consistently lower FID scores throughout training. The samples from the two fine-tuned models are shown in Figures 4 and 5. In comparison, the samples generated from the model fine-tuned by continuous-time RL have clearer contours, better aligned with real-world features, and exhibit superior aesthetic quality.

### 5.2. Fine-Tuning Stable Diffusion

**Setup**. We also validate our proposed algorithm for fine-tuning large-scale T2I diffusion models, Stable Diffusion

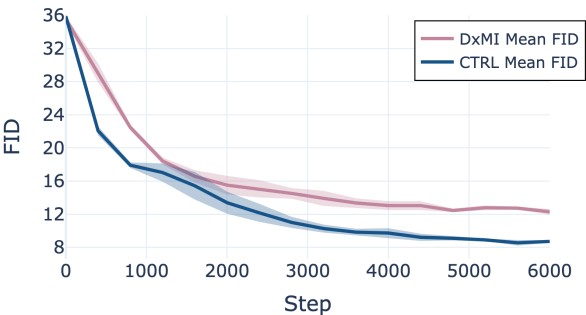

*Figure 3.* Training curves of DxMI and continuous-time RL.

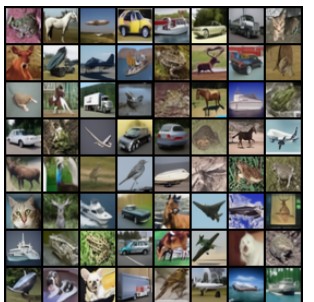 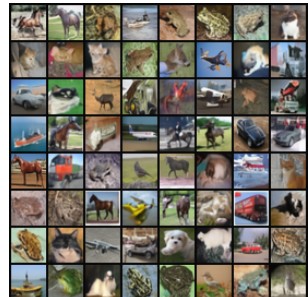

*Figure 4.* DxMI samples at the 6000-th step

*Figure 5.* Continuous-time RL samples at the 6000-th step

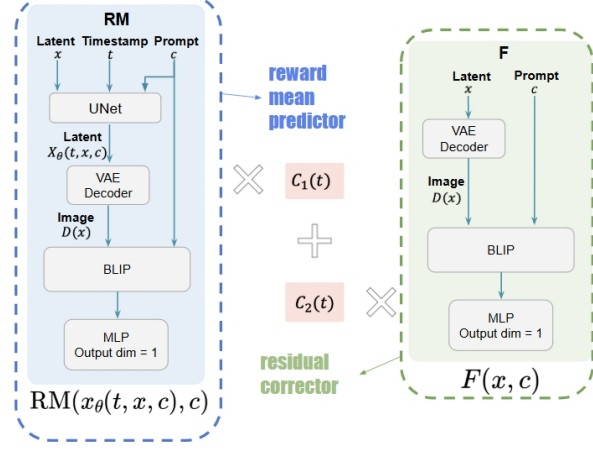

*Figure 6.* We adopt the similar backbone of ImageReward for two parts in value network, both by adding an MLP layer over the BLIP encoded latents.

v1.5 [3]. We adopt the pretrained ImageReward (Xu et al., 2024) as the reward signal during RL, as it has been shown in previous studies to achieve better alignment with human preferences to other metrics such as aesthetic scores, CLIP and BLIP scores.

---

[3]https://huggingface.co/stable-diffusion-v1-5/stable-diffusion-v1-5

We train the value networks with full parameter tuning, while we use LoRA (Hu et al., 2021) for tuning the U-nets of diffusion models. We adopt a learning rate of $10^{-7}$ for optimizing the value network, $3 \times 10^{-5}$ for optimizing the U-net and $\beta = 5 \times 10^{-5}$ for regularization. We train the models on 8 H200 GPUs with 128 effective batch sizes.

**Value Network Architecture.** Since we fix the reward model as ImageReward, we design the value network by using a similar backbone to the ImageReward model, which is composed of BLIP and a MLP header (see Figure 6). To ensure the boundary condition, we fix the parameters (i.e., BLIP and MLP) in the left part (skyblue) of the value network and only tune 30% of the parameters of BLIP in the right part (green). The VAE Decoder on both parts is fixed for efficiency and stabilized training.

As a remark, replacing $x$ with $x_\theta(t, x)$ in the "residual corrector" leads to minimum gain, compared to the drastic improvement brought forth by using $x_\theta(t, x)$ as the input in the "reward mean predictor". See Figure 2 and Table 1 for our ablation of network architecture and MSE statistics.

**Policies trained by Continuous-time RL are robust to time discretization**. We find that the policies trained by continuous-time RL achieve coherent performance in terms of the reward mean evaluated by ImageReward. In Figure 7, three line plots that correspond to 25, 50, and 100 steps almost always overlap after 20 epochs of training.

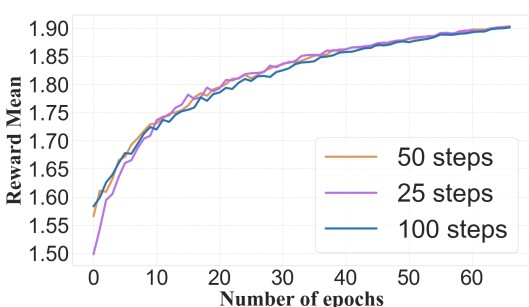

*Figure 7.* Performance of continuous-time RL's checkpoints with respect to discretization timesteps.

This showcases that our continuous-time RL trained policy is robust to time discretization, which is consistent with our theoretical analysis. Qualitative examples with the same prompt generated by the base model, checkpoints of 50 steps and 100 steps after continuous-time RL training can be found in Figure 8.

**Continuous-time RL outperforms Discrete-time RL baseline methods in both efficiency and stability**. We also compare the reward curves of discrete-time RL with our continuous-time RL algorithms. In Figure 9, the performance of the continuous-time RL is much more stable, and is more efficient in achieving a high average reward. We include the comparison with other baseline diffusion

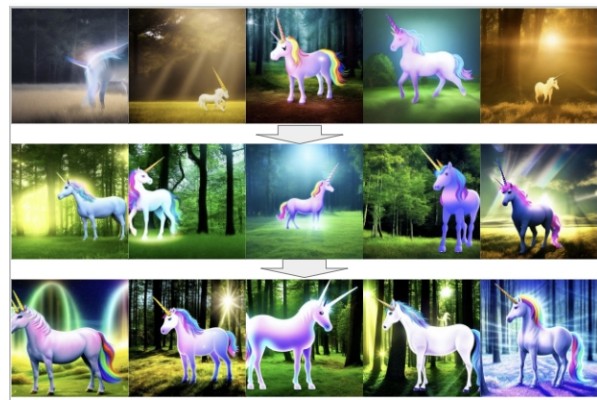

*Figure 8.* Model generations with prompt "A unicorn in a clearing. It has a single shining horn. Volumetric light." a) **Top:** Base model Stable Diffusion v1.5; b) **Mid:** Continuous-time RL after 50 training steps; c) **Bot:** Continuous-time RL after 100 training steps.

models fine-tuning methods like DRaFT (Clark et al., 2023) and AlignProp (Prabhudesai et al., 2023) in Appendix C.1.

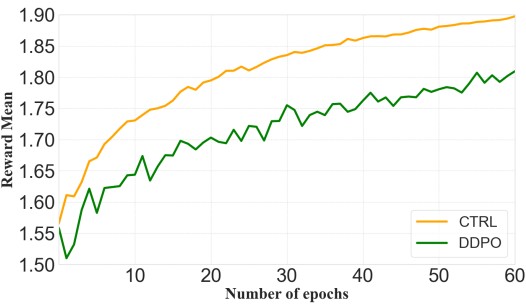

*Figure 9.* Performance of continuous-time RL against discrete-time RL under the same 50 discretization timesteps.

Why continuous-time approaches show better performance? Here we provide a heuristic explanation. Discrete-time RL methods optimize the objective with a priori time-discretization, which induces an error such that the resulting optimal policy can be significantly away from the true optimum in continuous time. Continuous-time RL methods, on the other hand, only require time-discretization in estimating the policy gradient. The error caused by this discretization — the gap between the resulting optimum and the true (continuous-time objective) optimum — is bounded by a polynomial of the step size (in gradient estimation) under suitable regularity conditions.

## 6. Discussion and Conclusion

We have proposed in this study a continuous-time reinforcement learning (RL) framework for fine-tuning diffusion models. Our work introduces novel policy optimization theory for RL in continuous time and space, alongside a

scalable and effective RL algorithm that enhances the generation quality of diffusion models, as validated by our experiments.

In addition, our algorithm and network designs exhibit a striking versatility that allows us to incorporate and leverage some of the advantages of prior works in diffusion models design, so as to better exploit model structures and to improve value network architectures. In view of this, we believe the continuous-time RL, in providing cross-pollination between diffusion models and RLHF, presents a highly promising direction for future research.

## Impact Statement

This paper presents work whose goal is to advance the field of Machine Learning. There are many potential societal consequences of our work, none which we feel must be specifically highlighted here.

## Acknowledgement

Hanyang Zhao, Haoxian Chen and Wenpin Tang are supported by NSF grant DMS-2113779 and DMS-2206038. Haoxian Chen is supported by the Amazon CAIT fellowship. Wenpin Tang receives support from the Columbia Innovation Hub grant, and the Tang Family Assistant Professorship. The works of Haoxian Chen, Hanyang Zhao, and David Yao are part of a Columbia-CityU/HK collaborative project that is supported by InnoHK Initiative, The Government of the HKSAR and the AIFT Lab.

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

# A. Connection between discrete-time and continuous-time sampler

In this section, we summarize the discussion of popular samplers like DDPM, DDIM, stochastic DDIM and their continuous-time limits being a Variance Preserving (VP) SDE.

## A.1. DDPM sampler is the discretization of VP-SDE

We review the forward and backward process in DDPM, and its connection to the VP SDE following the discussion in (Song et al., 2021b; Tang & Zhao, 2024). DDPM considers a sequence of positive noise scales $0 < \beta_1, \beta_2, \cdots, \beta_N < 1$. For each training data point $x_0 \sim p_{\text{data}}(x)$, a discrete Markov chain $\{x_0, x_1, \cdots, x_N\}$ is constructed such that:

$$x_i = \sqrt{1 - \beta_i} x_{i-1} + \sqrt{\beta_i} z_{i-1}, \quad i = 1, \cdots, N, \tag{28}$$

where $z_{i-1} \sim \mathcal{N}(0, I)$, thus $p(x_i \mid x_{i-1}) = \mathcal{N}(x_i; \sqrt{1 - \beta_i} x_{i-1}, \beta_i I)$. We can further think of $x_i$ as the $i^{\text{th}}$ point of a uniform discretization of time interval $[0, T]$ with discretization stepsize $\Delta t = \frac{T}{N}$, i.e. $x_{i\Delta t} = x_i$; and also $z_{i\Delta t} = z_i$. To obtain the limit of the Markov chain when $N \to \infty$, we define a function $\beta : [0, T] \to \mathbb{R}^+$ assuming that the limit exists: $\beta(t) = \lim_{\Delta t \to 0} \beta_i / \Delta t$ with $i = t/\Delta t$. Then when $\Delta t$ is small, we get:

$$x_{t+\Delta t} \approx \sqrt{1 - \beta(t)\Delta t} x_t + \sqrt{\beta(t)\Delta t} z_t \approx x_t - \frac{1}{2}\beta(t)x_t\Delta t + \sqrt{\beta(t)\Delta t} z_t.$$

Further taking the limit $\Delta t \to 0$, this leads to:

$$dX_t = -\frac{1}{2}\beta(t)X_t dt + \sqrt{\beta(t)} dB_t, \quad 0 \le t \le T,$$

and we have:

$$f(t, x) = -\frac{1}{2}\beta(t)x, g(t) = \sqrt{\beta(t)}.$$

Through reparameterization, we have $p_{\bar{\alpha}_i}(x_i \mid x_0) = \mathcal{N}(x_i; \sqrt{\bar{\alpha}_i} x_0, (1 - \bar{\alpha}_i) I)$, where $\bar{\alpha}_i := \prod_{j=1}^{i}(1 - \beta_j)$. For the backward process, a variational Markov chain in the reverse direction is parameterized with $p_\theta(x_{i-1} \mid x_i) = \mathcal{N}\left(x_{i-1}; \frac{1}{\sqrt{1 - \beta_i}}(x_i + \beta_i s_\theta(i, x_i)), \beta_i I\right)$, and trained with a re-weighted variant of the evidence lower bound (ELBO):

$$\theta^* = \arg\min_\theta \sum_{i=1}^{N} (1 - \bar{\alpha}_i) \mathbb{E}_{p_{\text{data}}(x)} \mathbb{E}_{p_{\bar{\alpha}_i}(\tilde{x}|x)} \left[ \|s_\theta(i, \tilde{x}) - \nabla_{\tilde{x}} \log p_{\bar{\alpha}_i}(\tilde{x} \mid x)\|_2^2 \right].$$

After getting the optimal model $s_{\theta^*}(i, x)$, samples can be generated by starting from $x_N \sim \mathcal{N}(0, I)$ and following the estimated reverse Markov chain as:

$$x_{i-1} = \frac{1}{\sqrt{1 - \beta_i}}(x_i + \beta_i s_{\theta^*}(i, x_i)) + \sqrt{\beta_i} z_i, \quad i = N, N-1, \cdots, 1. \tag{29}$$

Similar discussion as for the forward process, the equation (29) can further be rewritten as:

$$\begin{aligned}
x_{(i-1)\Delta t} &\approx \frac{1}{\sqrt{1 - \beta_{i\Delta t}\Delta t}}(x_{i\Delta t} + \beta(i\Delta t)\Delta t \cdot s_{\theta^*}(i\Delta t, x_{i\Delta t})) + \sqrt{\beta_i} z_i, \\
&\approx (1 + \frac{1}{2}\beta_{i\Delta t}\Delta t)(x_{i\Delta t} + \beta(i\Delta t)\Delta t \cdot s_{\theta^*}(i\Delta t, x_{i\Delta t})) + \sqrt{\beta_i} z_i, \\
&\approx (1 + \frac{1}{2}\beta_{i\Delta t}\Delta t)x_{i\Delta t} + \beta(i\Delta t)\Delta t \cdot s_{\theta^*}(i\Delta t, x_{i\Delta t}) + \sqrt{\beta_i} z_i,
\end{aligned} \tag{30}$$

when $\beta_{i\Delta t}$ is small. This is indeed the time discretization of the backward SDE:

$$\begin{aligned}
dX_t^\leftarrow &= (\frac{1}{2}\beta(T-t)X_t^\leftarrow + \beta(T-t)s_{\theta^*}(T-t, X_t^\leftarrow))dt + \sqrt{\beta(t)} dB_t, \\
&= \left(-f(T-t, X_t^\leftarrow) + g^2(T-t)s_{\theta^*}(T-t, X_t^\leftarrow)\right) dt + g(T-t) dB_t.
\end{aligned} \tag{31}$$

## A.2. DDIM sampler is the discretization of ODE

We review the backward process in DDIM, and its connection to the probability flow ODE following the discussion in (Song et al., 2021b; Kingma et al., 2021; Salimans & Ho, 2022; Zhang & Chen, 2022).

(i) DDIM update rule: The concrete updated rule in DDIM paper (same as in the implementation) adopted the following rule (with $\sigma_t = 0$ in Equation (12) of (Song et al., 2020)):

$$x_{t-1} = \sqrt{\bar{\alpha}_{t-1}} \underbrace{\left( \frac{x_t - \sqrt{1 - \bar{\alpha}_t} \epsilon_\theta^{(t)}(x_t)}{\sqrt{\bar{\alpha}_t}} \right)}_{\text{"predicted } x_0 \text{"}} + \underbrace{\sqrt{1 - \bar{\alpha}_{t-1}} \cdot \epsilon_\theta^{(t)}(x_t)}_{\text{"direction pointing to } x_t \text{"}} \tag{32}$$

To show the correspondence between DDIM parameters and continuous-time SDE parameters, we follow one derivation in (Salimans & Ho, 2022) by considering the "predicted $x_0$": note that define the predicted $x_0$ parameterization as:

$$\hat{x}_\theta(t, x) = \frac{x - \sqrt{1 - \bar{\alpha}_t} \epsilon_\theta^{(t)}(x)}{\sqrt{\bar{\alpha}_t}}, \text{ or }, \epsilon_\theta^{(t)}(x) = \frac{x - \sqrt{\bar{\alpha}_t} \hat{x}_\theta(t, x)}{\sqrt{1 - \bar{\alpha}_t}},$$

above (32) can be rewritten as:

$$x_{t-1} = \frac{\sqrt{1 - \bar{\alpha}_{t-1}}}{\sqrt{1 - \bar{\alpha}_t}} \left( x_t - \sqrt{\bar{\alpha}_t} \hat{x}_\theta(t, x) \right) + \sqrt{\bar{\alpha}_{t-1}} \cdot \hat{x}_\theta(t, x) \tag{33}$$

Using parameterization $\sigma_t = \sqrt{1 - \bar{\alpha}_t}$ and $\alpha_t = \sqrt{\bar{\alpha}_t}$, we have for $t - 1 = s < t$:

$$X_s = \frac{\sigma_s}{\sigma_t} \left[ X_t - \alpha_t \hat{x}_\theta(t, X_t) \right] + \alpha_s \hat{x}_\theta(t, X_t), \tag{34}$$

which is the same as derived in (Kingma et al., 2021; Salimans & Ho, 2022).

### A.2.1. ODE EXPLANATION BY ANALYZING THE DERIVATIVE

We further assume a VP diffusion process with $\alpha_t^2 = 1 - \sigma_t^2 = \text{sigmoid}(\lambda_t)$ for $\lambda_t = \log\left[\alpha_t^2 / \sigma_t^2\right]$, in which $\lambda_t$ is known as the signal-to-noise ratio. Taking the derivative of (34) with respect to $\lambda_s$, assuming again a variance preserving diffusion process, and using $\frac{d\alpha_\lambda}{d\lambda} = \frac{1}{2} \alpha_\lambda \sigma_\lambda^2$ and $\frac{d\sigma_\lambda}{d\lambda} = -\frac{1}{2} \sigma_\lambda \alpha_\lambda^2$, gives

$$\begin{aligned}
\frac{X_{\lambda_s}}{d\lambda_s} &= \frac{d\sigma_{\lambda_s}}{d\lambda_s} \frac{1}{\sigma_t} \left[ X_t - \alpha_t \hat{x}_\theta(t, X_t) \right] + \frac{d\alpha_{\lambda_s}}{d\lambda_s} \hat{x}_\theta(t, X_t) \\
&= -\frac{1}{2} \alpha_s^2 \frac{\sigma_s}{\sigma_t} \left[ X_t - \alpha_t \hat{x}_\theta(t, X_t) \right] + \frac{1}{2} \alpha_s \sigma_s^2 \hat{x}_\theta(t, X_t).
\end{aligned}$$

Evaluating this derivative at $s = t$ then gives

$$\begin{aligned}
\left. \frac{X_{\lambda_s}}{d\lambda_s} \right|_{s=t} &= -\frac{1}{2} \alpha_\lambda^2 \left[ X_\lambda - \alpha_\lambda \hat{x}_\theta(t, X_\lambda) \right] + \frac{1}{2} \alpha_\lambda \sigma_\lambda^2 \hat{x}_\theta(t, X_\lambda) \\
&= -\frac{1}{2} \alpha_\lambda^2 \left[ X_\lambda - \alpha_\lambda \hat{x}_\theta(t, X_\lambda) \right] + \frac{1}{2} \alpha_\lambda \left( 1 - \alpha_\lambda^2 \right) \hat{x}_\theta(t, X_\lambda) \\
&= \frac{1}{2} \left[ \alpha_\lambda \hat{x}_\theta(t, X_\lambda) - \alpha_\lambda^2 X_\lambda \right].
\end{aligned} \tag{35}$$

Recall that the forward process in terms of an SDE is defined as:

$$dX_t = f(t, X_t)dt + g(t)dB_t, \quad t \in [0, T]$$

and (Song et al., 2021b) shows that backward of this diffusion process is an SDE, but shares the same marginal probability density of an associated probability flow ODE (by taking $t := T - t$):

$$dX_t = \left[ f(t, X_t) - \frac{1}{2} g^2(t) \nabla_x \log p(t, X_t) \right] dt, \quad t \in [T, 0]$$

where in practice $\nabla_x \log p(t, x)$ is approximated by a learned denoising model using

$$\nabla_x \log p(t, x) \approx s_\theta(t, x) = \frac{\alpha_t \hat{x}_\theta (t, x) - x}{\sigma_t^2} = -\frac{\epsilon_\theta^{(t)} (x)}{\sigma_t}. \tag{36}$$

with two chosen noise scheduling parameters $\alpha_t$ and $\sigma_t$, and corresponding drift term $f(t, x) = \frac{d \log \alpha_t}{dt} x_t$ and diffusion term $g^2(t) = \frac{d\sigma_t^2}{dt} - 2\frac{d \log \alpha_t}{dt}\sigma_t^2$.

Further assuming a VP diffusion process with $\alpha_t^2 = 1 - \sigma_t^2 = \text{sigmoid}(\lambda_t)$ for $\lambda_t = \log [\alpha_t^2/\sigma_t^2]$, we get

$$f(t, x) = \frac{d \log \alpha_t}{dt} x = \frac{1}{2}\frac{d \log \alpha_\lambda^2}{d\lambda}\frac{d\lambda}{dt} x = \frac{1}{2}\left(1 - \alpha_t^2\right)\frac{d\lambda}{dt} x = \frac{1}{2}\sigma_t^2\frac{d\lambda}{dt} x.$$

Similarly, we get

$$g^2(t) = \frac{d\sigma_t^2}{dt} - 2\frac{d \log \alpha_t}{dt}\sigma_t^2 = \frac{d\sigma_\lambda^2}{d\lambda}\frac{d\lambda}{dt} - \sigma_t^4\frac{d\lambda}{dt} = \left(\sigma_t^4 - \sigma_t^2\right)\frac{d\lambda}{dt} - \sigma_t^4\frac{d\lambda}{dt} = -\sigma_t^2\frac{d\lambda}{dt}.$$

Plugging these into the probability flow ODE then gives

$$\begin{aligned}
\mathrm{d}X_t &= \left[f(t, X_t) - \frac{1}{2}g^2(t)\nabla_x \log p(t, x)\right]\mathrm{d}t \\
&= \frac{1}{2}\sigma_t^2\left[X_t + \nabla_x \log p(t, X_t)\right]\mathrm{d}\lambda_t.
\end{aligned} \tag{37}$$

Plugging in our function approximation from Equation (36) gives

$$\begin{aligned}
\mathrm{d}X_t &= \frac{1}{2}\sigma_t^2\left[X_t + \left(\frac{\alpha_t \hat{x}_\theta (t, X_t) - X_t}{\sigma_t^2}\right)\right]\mathrm{d}\lambda_t \\
&= \frac{1}{2}\left[\alpha_t \hat{x}_\theta (t, X_t) + \left(\sigma_t^2 - 1\right) X_t\right]\mathrm{d}\lambda_t \\
&= \frac{1}{2}\left[\alpha_t \hat{x}_\theta (t, X_t) - \alpha_t^2 X_t\right]\mathrm{d}\lambda_t.
\end{aligned} \tag{38}$$

Comparison this with Equation (35) now shows that DDIM follows the probability flow ODE up to first order, and can thus be considered as an integration rule for this ODE.

### A.2.2. EXPONENTIAL INTEGRATOR EXPLANATION

In (Zhang & Chen, 2022) that the integration role above is referred as "exponential integrator" of (38). We adopt two ways of derivations:

(a) Notice that, if we treat the $\hat{x}_\theta (t, X_t)$ as a constant in (38) (or assume that it does not change w.r.p. $t$ along the ODE trajectory), we have:

$$\mathrm{d}X_t + \frac{1}{2}\alpha_t^2 X_t \mathrm{d}\lambda_t = \hat{x}_\theta (t, X_t) \cdot \frac{1}{2}\alpha_t \mathrm{d}\lambda_t. \tag{39}$$

Both sides multiplied by $1/\sigma_t$ and integrate from $t$ to $s$ yields:

$$\frac{X_s}{\sigma_s} - \frac{X_t}{\sigma_t} = \hat{x}_\theta (t, X_t) \cdot \left(\exp(\frac{1}{2}\lambda_s) - \exp(\frac{1}{2}\lambda_t)\right) = \hat{x}_\theta (t, X_t) \cdot \left(\frac{\alpha_s}{\sigma_s} - \frac{\alpha_t}{\sigma_t}\right). \tag{40}$$

which is thus

$$X_s = \frac{\sigma_s}{\sigma_t}X_t + \left[\alpha_s - \alpha_t\frac{\sigma_s}{\sigma_t}\right]\hat{x}_\theta (t, X_t), \tag{41}$$

which is the same as DDIM continuous-time intepretation as in (34).

(b) We also notice that we can also simplify the whole proof by treating the scaled score (same as in (Zhang & Chen, 2022)):

$$\sigma_t \nabla_x \log p(t, x) \approx \sigma_t s_\theta(t, x) = \frac{\alpha_t \hat{x}_\theta (t, x) - x}{\sigma_t} \tag{42}$$

as a constant in (37) (or assume that it does not change w.r.p. $t$ along the ODE trajectory). Notice that from backward ODE, we have:

$$\mathrm{d}X_t = \frac{1}{2}\sigma_t^2 \left[ X_t + \frac{1}{\sigma_t}\sigma_t \nabla_x \log p(t, X_t) \right] \mathrm{d}\lambda_t. \tag{43}$$

Both sides multiplied by $1/\alpha_t$ and integrate from $t$ to $s$ yields:

$$\frac{X_s}{\alpha_s} - \frac{X_t}{\alpha_t} = \left( \frac{\alpha_t \hat{x}_\theta(t, X_t) - X_t}{\sigma_t} \right) \cdot \left( -\frac{\sigma_s}{\alpha_s} + \frac{\sigma_t}{\alpha_t} \right). \tag{44}$$

which is thus

$$X_s = \frac{\sigma_s}{\sigma_t} X_t + \left[ \alpha_s - \alpha_t \frac{\sigma_s}{\sigma_t} \right] \hat{x}_\theta(t, X_t), \tag{45}$$

which is the same as DDIM continuous-time intepretation as in (34).

As a summary, treating the denoised mean or the noise predictor as the constants will both recovery the rule of DDIM. Usually, for ODE flows, the denoised mean assumption naturally holds; however, why the scaled score leads to the same integration rule remains to be an interesting question, probably comes from the design property of DDIM, see e.g. discussions in (Karras et al., 2022).

# B. Theorem Proofs

## B.1. Proof of Theorem 3.1

The main proof technique relies on Girsanov's Theorem, which is similar to the argument in (Chen et al., 2022). First, we recall a consequence of Girsanov's theorem that can be obtained by combining Pages 136-139, Theorem 5.22, and Theorem 4.13 of Le Gall (2016).

**Theorem B.1.** *For $t \in [0, T]$, let $\mathcal{L}_t = \int_0^t b_s \, \mathrm{d}B_s$ where $B$ is a $Q$-Brownian motion. Assume that $\mathbb{E}_Q \int_0^T \|b_s\|^2 \, \mathrm{d}s < \infty$. Then, $\mathcal{L}$ is a $Q$-martingale in $L^2(Q)$. Moreover, if*

$$\mathbb{E}_Q \mathcal{E}(\mathcal{L})_T = 1, \quad \text{where } \mathcal{E}(\mathcal{L})_t := \exp\left( \int_0^t b_s \, \mathrm{d}B_s - \frac{1}{2} \int_0^t \|b_s\|^2 \, \mathrm{d}s \right), \tag{46}$$

*then $\mathcal{E}(\mathcal{L})$ is also a $Q$-martingale, and the process*

$$t \mapsto B_t - \int_0^t b_s \, \mathrm{d}s \tag{47}$$

*is a Brownian motion under $P := \mathcal{E}(\mathcal{L})_T Q$, the probability distribution with density $\mathcal{E}(\mathcal{L})_T$ w.r.t. $Q$.*

If the assumptions of Girsanov's theorem are satisfied (i.e., the condition (46)), we can apply Girsanov's theorem to $Q$ as the law of the following reverse process (we omit $c$ for brevity),

$$\mathrm{d}\overline{X}_t = \left( -f(T-t, \overline{X}_t) + g^2(T-t)s_{\theta_{pre}}(T-t, \overline{X}_t) \right) \mathrm{d}t + g(T-t)\mathrm{d}B_t, \quad \overline{X}_0 \sim p_\infty(\cdot) \tag{48}$$

and

$$b_t = g(T-t) \left[ s_\theta(T-t, \overline{X}_t) - s_{\theta_{pre}}(T-t, \overline{X}_t) \right], \tag{49}$$

where $t \in [0, T]$. This tells us that under $P = \mathcal{E}(\mathcal{L})_T Q$, there exists a Brownian motion $(\beta_t)_{t \in [0,T]}$ s.t.

$$\mathrm{d}B_t = g(T-t) \left[ s_\theta(T-t, \overline{X}_t) - s_{\theta_{pre}}(T-t, \overline{X}_t) \right] \mathrm{d}t + \mathrm{d}\beta_t. \tag{50}$$

Plugging (50) into (48) we have $P$-a.s.,

$$\mathrm{d}\overline{X}_t = \left( -f(T-t, \overline{X}_t) + g^2(T-t)s_\theta(T-t, \overline{X}_t) \right) \mathrm{d}t + g(T-t)\mathrm{d}\beta_t, \quad \overline{X}_0 \sim p_\infty(\cdot) \tag{51}$$

In other words, under $P$, the distribution of $\overline{X}$ is the same as the distribution generated by current policy parameterized by $\theta$, i.e., $p_\theta(\cdot) = P_T = \mathcal{E}(\mathcal{L})_T Q$. Therefore,

$$
\begin{aligned}
D_{KL}\left(p_\theta \| p_{\theta_{pre}}\right) &= \mathbb{E}_{P_T} \ln \frac{\mathrm{d}P_T}{\mathrm{d}Q_T} = \mathbb{E}_{P_T} \ln \mathcal{E}(\mathcal{L})_T \\
&= \mathbb{E}_{P_T}\left[\int_0^t b_s \, \mathrm{d}B_s - \frac{1}{2}\int_0^t \|b_s\|^2\right] \\
&= \mathbb{E}_{P_T}\left[\int_0^t b_s \, \mathrm{d}\beta_s + \frac{1}{2}\int_0^t \|b_s\|^2\right] \\
&= \frac{1}{2}\int_0^t g^2(T-t)\underbrace{\mathbb{E}_P \left\| s_\theta(T-t, \overline{X}_t) - s_{\theta_{pre}}(T-t, \overline{X}_t)\right\|^2}_{\epsilon_t^2} \mathrm{d}t
\end{aligned}
$$

Thus we can bound the discrepancy between distribution generated by the policy $\theta$ and the pretrained parameters $\theta_{pre}$ as

$$
D_{KL}(p_\theta \| p_{\theta_{pre}}) \leq \frac{1}{2}\int_0^T g^2(T-t)\epsilon_t^2 \mathrm{d}t \tag{52}
$$

### B.2. Proof of Theorem 4.1

First we include the policy gradient formula theorem for finite horizon in continuous time from (Jia & Zhou, 2022b):

**Lemma B.2** (Theorem 5 of (Jia & Zhou, 2022b) when $R \equiv 0$). *Under some regularity conditions, given an admissible parameterized policy $\pi_\theta$, the policy gradient of the value function $V\left(t, x; \pi^\theta\right)$ admits the following representation:*

$$
\begin{aligned}
\frac{\partial}{\partial \theta}V(t, x; \pi^\theta) =& \mathbb{E}^{\mathbb{P}}\left[\int_t^T e^{-\beta(s-t)}\left\{\frac{\partial}{\partial \theta}\log \pi^\theta(a_s^{\pi^\theta}|s, X_s^{\pi^\theta})\left(\mathrm{d}V(s, X_s^{\pi^\theta}; \pi^\theta)\right.\right.\right. \\
&\left.\left.\left.+ \left[r_R(s, X_s^{\pi^\theta}, a_s^{\pi^\theta}) - \beta V(s, X_s^{\pi^\theta}; \pi^\theta)\right] \mathrm{d}s\right)\right\} \mid X_t^{\pi^\theta} = x\right], \quad (t, x) \in [0, T] \times \mathbb{R}^d
\end{aligned} \tag{53}
$$

*in which we denote the regularized reward*

$$
r_R(t, X_t^{\pi^\theta}, a_t^{\pi^\theta}) = \gamma(t)\|a_t^{\pi^\theta} - s^{\theta^*}(t, X_t)\|^2.
$$

First, by applying Itô's formula to $V(t, X_t)$, we have:

$$
\mathrm{d}V(t, X_t) = \left[\frac{\partial V}{\partial t}(t, X_t) + \frac{1}{2}\sigma(t)^2 \circ \frac{\partial^2 V}{\partial x^2}(t, X_t)\right]\mathrm{d}t + \frac{\partial V}{\partial x}(t, X_t)\mathrm{d}X_t. \tag{54}
$$

Further recall that:

$$
q(t, x, a; \pi) = \frac{\partial V}{\partial t}(t, x; \pi) + \mathcal{H}\left(t, x, a, \frac{\partial V}{\partial x}(t, x; \pi), \frac{\partial^2 V}{\partial x^2}(t, x; \pi)\right) - \beta V(t, x; \pi), \tag{55}
$$

this implies that (similar discussion also appeared in (Jia & Zhou, 2023))

$$
q\left(t, X_t^\pi, a_t^\pi; \pi\right)\mathrm{d}t = \mathrm{d}J\left(t, X_t^\pi; \pi\right) + r\left(t, X_t^\pi, a_t^\pi\right)\mathrm{d}t - \beta J\left(t, X_t^\pi; \pi\right)\mathrm{d}t + \{\cdots\}\mathrm{d}B_t. \tag{56}
$$

Plug this equality back in (53) yields:

$$
\frac{\partial}{\partial \theta}V(t, x; \pi^\theta) = \mathbb{E}^{\mathbb{P}}\left[\int_t^T e^{-\beta(s-t)}\frac{\partial}{\partial \theta}\log \pi^\theta(a_s^{\pi^\theta}|s, X_s^{\pi^\theta})q\left(t, X_t^\pi, a_t^\pi; \pi\right)\mathrm{d}s \mid X_t^{\pi^\theta} = x\right], \tag{57}
$$

Let $t = 0$, $\beta = -\alpha$ and further taking expectation to the initial distribution yields Theorem 4.1.

## B.3. Proof of Theorem 4.4

We recall the definition of terms and rewrite the bound in Theorem 4.4 for convenience. It suffices to prove that there exists constant $C_1$ and $C_2$, such that

$$|V^{\hat{\theta}} - V^{\theta}| \leq C_1 \cdot \left( \mathbb{E} \int_0^T \mathrm{KL}(\pi^{\theta}(\cdot|t, X_t^{\theta})\|\pi^{\hat{\theta}}(\cdot|t, X_t^{\theta}))\mathrm{d}t \right)^{\frac{1}{2}}, \tag{58}$$

and

$$|L^{\theta}(\hat{\theta}) - V^{\theta}| \leq C_2 \cdot \left( \mathbb{E} \int_0^T \mathrm{KL}(\pi^{\theta}(\cdot|t, X_t^{\theta})\|\pi^{\hat{\theta}}(\cdot|t, X_t^{\theta}))\mathrm{d}t \right)^{\frac{1}{2}}. \tag{59}$$

To prove (58), we need to utilize the KL-regularized objective we study for this paper, as this might not hold under general cases. Explicitly writing the definition of $V^{\theta}$ and $V^{\hat{\theta}}$, we have (excluding contents $c$ for simplicity):

$$|V^{\hat{\theta}} - V^{\theta}| \leq \underbrace{|\mathbb{E}(\mathrm{RM}(X_T^{\hat{\theta}})) - \mathbb{E}(\mathrm{RM}(X_T^{\theta}))|}_{\text{(i) reward difference}} + \beta \underbrace{|\mathrm{KL}\left(p^{\hat{\theta}}(T, \cdot)\|p^{\theta_{pre}}(T, \cdot)\right) - \mathrm{KL}\left(p^{\theta}(T, \cdot)\|p^{\theta_{pre}}(T, \cdot)\right)|}_{\text{(ii) KL difference}}.$$

For bounding (i), we have:

$$\text{(i)} = \int_x (p^{\hat{\theta}}(T, x) - p^{\theta}(T, x))\mathrm{RM}(x)\mathrm{d}x| \leq |\int_x |p^{\hat{\theta}}(T, x) - p^{\theta}(T, x)||\mathrm{RM}(x)|\mathrm{d}x \leq N \cdot \sqrt{\mathrm{KL}\left(p^{\theta}(T, \cdot)\|p^{\hat{\theta}}(T, \cdot)\right)},$$

where the last equality is due to Pinsker's inequality. For bounding (ii), a standard identity for Kullback–Leibler divergences gives

$$\mathrm{KL}\big(p^{\hat{\theta}} \,\|\, p^{\theta_{\mathrm{pre}}}\big) \;-\; \mathrm{KL}\big(p^{\theta} \,\|\, p^{\theta_{\mathrm{pre}}}\big) \;=\; -\mathrm{KL}\big(p^{\theta} \,\|\, p^{\hat{\theta}}\big) \;+\; \int \big[p^{\hat{\theta}}(x) \;-\; p^{\theta}(x)\big] \ln\!\Big(\tfrac{p^{\hat{\theta}}(x)}{p^{\theta_{\mathrm{pre}}}(x)}\Big) \,\mathrm{d}x.$$

Hence, by the triangle inequality,

$$\left|\mathrm{KL}\big(p^{\hat{\theta}} \,\|\, p^{\theta_{\mathrm{pre}}}\big) - \mathrm{KL}\big(p^{\theta} \,\|\, p^{\theta_{\mathrm{pre}}}\big)\right| \;\leq\; \mathrm{KL}\big(p^{\theta} \,\|\, p^{\hat{\theta}}\big) \;+\; \left|\int (p^{\hat{\theta}} - p^{\theta}) \ln\!\big(\tfrac{p^{\hat{\theta}}}{p^{\theta_{\mathrm{pre}}}}\big)\right|.$$

By the bounded log-ratio assumption,

$$\left|\ln\!\big(\tfrac{p^{\hat{\theta}}(x)}{p^{\theta_{\mathrm{pre}}}(x)}\big)\right| \;\leq\; C_3,$$

so

$$\left|\int (p^{\hat{\theta}} - p^{\theta}) \ln\!\big(\tfrac{p^{\hat{\theta}}}{p^{\theta_{\mathrm{pre}}}}\big)\right| \;\leq\; C_3 \, \|p^{\hat{\theta}} - p^{\theta}\|_1.$$

Finally, Pinsker's inequality implies $\|p^{\hat{\theta}} - p^{\theta}\|_1 \leq \sqrt{2\,\mathrm{KL}\big(p^{\theta} \,\|\, p^{\hat{\theta}}\big)}$. Combining these bounds yields

$$\left|\mathrm{KL}\big(p^{\hat{\theta}} \,\|\, p^{\theta_{\mathrm{pre}}}\big) - \mathrm{KL}\big(p^{\theta} \,\|\, p^{\theta_{\mathrm{pre}}}\big)\right| \;\leq\; \mathrm{KL}\big(p^{\theta} \,\|\, p^{\hat{\theta}}\big) \;+\; C_4 \sqrt{2\,\mathrm{KL}\big(p^{\theta} \,\|\, p^{\hat{\theta}}\big)}.$$

Further given the assumption that $p^{\hat{\theta}}$ is close to $p^{\theta}$ such that $\mathrm{KL}(p^{\theta} \,\|\, p^{\hat{\theta}}) \leq 1$, we have

$$|V^{\hat{\theta}} - V^{\theta}| \leq C_5 \cdot \sqrt{\mathrm{KL}(p^{\theta} \,\|\, p^{\hat{\theta}})}.$$

Further applying the same argument in Theorem 3.1 proves (58). To prove (59), it suffices to bound

$$\mathbb{E} \int_0^T \frac{\pi^{\hat{\theta}}(a_t^{\theta}|t, X_t^{\theta})}{\pi^{\theta}(a_t^{\theta}|t, X_t^{\theta})} q(t, X_t^{\theta}, a_t^{\theta}; \pi^{\theta})\mathrm{d}t \tag{60}$$

By importance sampling, we have

$$\mathbb{E}_\theta\Big[\frac{\pi^{\hat{\theta}}(a_t^\theta \mid t, X_t^\theta)}{\pi^\theta(a_t^\theta \mid t, X_t^\theta)}\, q(t, X_t^\theta, a_t^\theta; \pi^\theta)\Big] \;=\; \mathbb{E}_{\hat{\theta}}[\, q(t, X_t^\theta, a_t^{\hat{\theta}}; \pi^\theta)\,].$$

Hence the left side equals $\int_0^T \mathbb{E}_{\hat{\theta}}[\, q(\cdots)\,]\mathrm{d}t$. Assume that $|q| \le M$, we get

$$\big|\mathbb{E}_{\hat{\theta}}[q] - \mathbb{E}_\theta[q]\big| \;\le\; M\,\|\pi^{\hat{\theta}} - \pi^\theta\|_1 \;\le\; M\,\sqrt{2\,\mathrm{KL}(\pi^\theta \,\|\, \pi^{\hat{\theta}})},$$

by Pinsker's inequality. Therefore,

$$\mathbb{E}_{\hat{\theta}}[q(\cdots)] \;\le\; \mathbb{E}_\theta[q(\cdots)] \;+\; M\,\sqrt{2\,\mathrm{KL}(\pi^\theta \,\|\, \pi^{\hat{\theta}})}.$$

Integrate over $t$ from $0$ to $T$, and the result follows that:

$$|L^\theta(\hat{\theta}) - V^\theta| \le C_6 \cdot \mathbb{E}\int_0^T \sqrt{\mathrm{KL}(\pi^\theta(\cdot|t, X_t^\theta)\|\pi^{\hat{\theta}}(\cdot|t, X_t^\theta))}\mathrm{d}t \le C_7\left(\mathbb{E}\int_0^T \mathrm{KL}(\pi^\theta(\cdot|t, X_t^\theta)\|\pi^{\hat{\theta}}(\cdot|t, X_t^\theta))\mathrm{d}t\right)^{\frac{1}{2}} \quad (61)$$

where the last inequality follows from Jensen's ineqality and Cauchy-Schwarz inequality.

## C. More Experimental Details

### C.1. Comparison with other discrete-time baselines for diffusion models fine-tuning

We also include the comparison with non RL based discrete time diffusion models fine-tuning baselines, DRaFT and AlignProp. The implementation of DRaFT is modified from AlignProp given their similarity. As in Figure 10, our CTRL method outperforms our discrete time baselines.

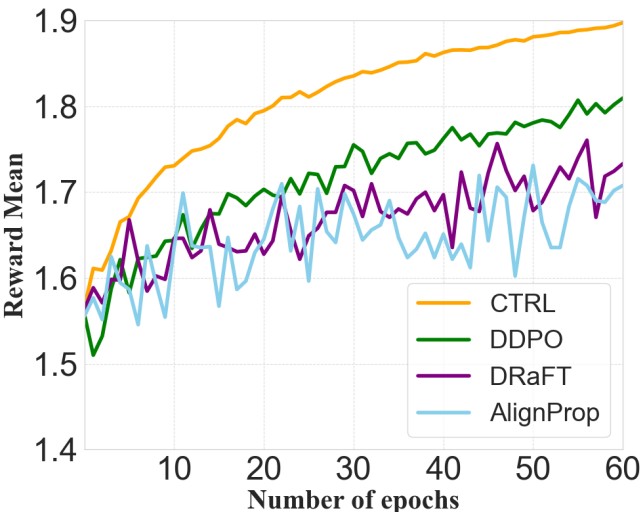

*Figure 10.* Performance of continuous-time RL against discrete-time baselines under the same 50 discretization timesteps.

### C.2. Algorithm Pseudocode

We present the algorithm pseudocode as below.

---

**Algorithm 1** CTRL for Diffusion Models

---

**Input:** Initial policy parameters $\theta_0$, value function parameters $\phi_0$ (after pretraining), clip parameter $\epsilon$, learning rates $\alpha_\theta$, $\alpha_\phi$, number of epochs $K$, number of trajectories $N$, a fixed exploration level $\sigma$, number of exploration actions $M$, scaling parameter $\eta$.

**for** $n = 0, 1, 2, \ldots$ **do**

    Collect $N$ trajectories $\{(t_i, X_{t_i}^{\theta_n}, a_{t_i}^{\theta_n}, r_{t_i}^{\theta_n})\}_{i=1}^N$ by running policy $\pi^{\theta_n}$

    Compute returns $\hat{R}_i = \sum_{t=t_i}^T r_t^{\theta_n}$ for each trajectory

    Initialize value function dataset $\mathcal{D}_V = \{(t_i, X_{t_i}^{\theta_n}, \hat{R}_i)\}$

    *# Train value function*

    **for** $k = 1, 2, \ldots, K$ **do**

        Sample batch $\mathcal{B}_V \subset \mathcal{D}_V$

        Update $\phi_{n+1} \leftarrow \phi_n - \alpha_\phi \nabla_\phi \frac{1}{|\mathcal{B}_V|} \sum_{(t, X_t, \hat{R}) \in \mathcal{B}_V} (V^\phi(t, X_t) - \hat{R})^2$

    **end for**

    *# Compute advantage rate function estimates*

    **for** each $(t_i, X_{t_i}^{\theta_n}, a_{t_i}^{\theta_n})$ in collected trajectories **do**

        Sample $M$ samples of random noise $\epsilon_j \sim \mathcal{N}(0, I)$.

        Compute $M$ pseudo samples $a_{t_i,j}^{\theta_n} = a_{t_i}^{\theta_n} + \sigma \epsilon_j$.

        Compute advantage $q_{t_i,j}^{\theta_n} = \left( V(t_i, X_{t_i}^{\theta_n} + \eta\, g^2(T-t)\epsilon_j) - V(t_i, X_{t_i}^{\theta_n}) \right) / \eta$

    **end for**

    Initialize policy optimization dataset $\mathcal{D}_\pi = \{(t_i, X_{t_i}^{\theta_n}, a_{t_i,j}^{\theta_n}, q_{t_i,j}^{\theta_n})\}$

    *# Update policy using PPO objective*

    **for** $k = 1, 2, \ldots, K$ **do**

        Sample batch $\mathcal{B}_\pi \subset \mathcal{D}_\pi$

        Compute likelihood ratios $\rho_{t,j}^\theta = \frac{\pi^\theta(a_{t,j}^{\theta_n}|t, X_t^{\theta_n})}{\pi^{\theta_n}(a_{t,j}^{\theta_n}|t, X_t^{\theta_n})}$ for all $(t, X_t^{\theta_n}, a_{t,j}^{\theta_n}, q_{t,j}^{\theta_n}) \in \mathcal{B}_\pi$

        Compute clipped objective:

        $L(\theta) = \frac{1}{|\mathcal{B}_\pi|} \sum_{(t, X_t, a_{t,j}, q_{t,j}) \in \mathcal{B}_\pi} \min(\rho_{t,j}^\theta q_{t,j}^{\theta_n}, \text{clip}(\rho_{t,j}^\theta, 1-\epsilon, 1+\epsilon) q_{t,j}^{\theta_n})$

        Update $\theta \leftarrow \theta + \alpha_\theta \nabla_\theta L(\theta)$

    **end for**

    $\theta_{n+1} \leftarrow \theta$

**end for**

---

