# OpenReview forum: "Score as Action: Fine Tuning Diffusion Generative Models by Continuous-time Reinforcement Learning"
_ICML.cc/2025/Conference — ICML 2025 poster_

### Official Review · Reviewer_HDpz · 2025-03-01

**Overall Recommendation:** 3

**Summary:**

The paper proposes a continuous-time reinforcement learning framework to fine-tune diffusion models by treating score functions as actions in a stochastic control problem. The authors derive a policy gradient theorem for continuous-time RL and connects KL regularization to a tractable running reward. The authors also develop a policy optimization theory by adapting ideas from TRPO/PPO to the continuous setting.

## update after rebuttal
Thanks for the response. I will keep my score.

**Claims And Evidence:**

The authors claim that the proposed continuous-time RL framework, specialized value function design and policy optimization algorithm can improve the quality of diffusion generative model’s output. Empirical results on both small-scale (CIFAR-10) and large-scale (Stable Diffusion v1.5) settings provide convincing evidence of the method’s effectiveness.

**Essential References Not Discussed:**

There is no essential reference not mentioned.

**Experimental Designs Or Analyses:**

The experiment design is mostly reasonable, but it would be better to elaborate more on the advantages of their policy optimization algorithm.

**Methods And Evaluation Criteria:**

YES

**Other Comments Or Suggestions:**

Nil

**Other Strengths And Weaknesses:**

Strengths:
1.	The idea of treating scoring function as action and aligning diffusion model fine-tuning with continuous-time RL is novel.
2.	The theoretical derivations provide a robust foundation for the proposed method and help to connect with the continuous framework.

Weaknesses:
1.	There is limited discussion on the additional computational cost that might arise from adopting a continuous-time formulation compared to conventional discrete-time methods.

**Questions For Authors:**

1.	Could you please provide some intuitive explanations for the value function design in Equation (20)?
2.	The authors claim that they provide a tighter bound and closed-form advantage-rate function compared with conventional continuous-time PPO approaches. How does this help to generate images with higher quality?

**Relation To Broader Scientific Literature:**

Most of the prior works of leveraging RL to fine-tune diffusion generative models use discrete-time formulation. The paper’s key idea of connecting diffusion generative models to continuous-time reinforcement learning is novel.

**Theoretical Claims:**

The authors claim that their continuous-time policy optimization leads to a tighter bound and a closed-form advantage-rate function compared with conventional continuous-time PPO approaches. There is no obvious error in the proofs.

---

> ### Author Rebuttal · Authors · 2025-03-31
>
> We sincerely thank you for providing thoughtful suggestions to improve our paper. Please find our responses to your questions below:
>
>  ***Q1: intuitive explanations for the value function design in Equation (20)***
>
> ***A1***: The design of our value function in Equation (20) is motivated to capture the structural property of score-based diffusion models (we explain in (a) below) and to meet the boundary condition of the value function (we explain in (b) below).
>
> - **(a)** Recall that the value function $V(t,x)$ is defined as the expected reward of final generated image (if regularization penalty parameter $\beta$ is 0) conditional on the current time $r$ and state/latent $x$, so an natural and naive approximation of it can be the reward of the denoised image $\hat{x}_\theta(t,x)$ of the current state $x$, as the denoised image can be understood as "a prediction of the image at the end of SDE trajectory". This illustrate our first term reward mean predictor; of course such approximation to the value function is not accurate, so we use the second term with the same network architecture to further approximate the possible residual term which can improve the performance by reducing the mean square error.
>
> - **(b)** The value function has a boundary condition that at the final time $T$, $V(T,x)=R(x)$, so we choose two proper coefficient functions $c_{\text{skip}}(t)$ and $c_{\text{out}}(t)$ to satisfy such needs: we let $c_{\text{skip}}(T)=1$ and $c_{\text{out}}(T)=0$, thus the boundary condition is naturally satisfied.
>
> We hope that this clarifies our design space and we are happy to take further questions.
>
> ***Q2: How does the closed-form advantage-rate function to generate images with higher quality?***
>
> ***A2***: We illustrate this mainly through the RL prospective as images with higher quality can be reflected as higher average reward for RL. As far as we know, the most relevant work about the conventional continuous-time PPO approaches is in [1], which estimates the advantage rate function through a model-free manner, but requires sampling many more examples to make the estimation accurate and algorithms stable. In our method, thanks to the closed-form advantage-rate function and further subtraction of second order term (which is hard to compute) as the baseline function (please find more detailed discussion about this in Line 309-322 after Theorem 1), we get a much simpler way to compute the advantage function which is also much more sample efficient, since we do not need to query the generative model for the next state to go. We believe the leverage of such structural property is one of the key reasons we get quite stable training reward curves as in illustrate in orange curve of Figure 9.
>
> [1]. Zhao et al. 2023. Policy optimization for continuous reinforcement learning. NeurIPS 2023

---

### Official Review · Reviewer_2RBC · 2025-03-11

**Overall Recommendation:** 4

**Summary:**

The paper proposes a continuous-time RL approach for fine-tuning diffusion generative models. It develops a policy optimization framework tailored for continuous-time RL, and empirically validate on the Stable Diffusion v1.5 model.

**Claims And Evidence:**

1. Most importantly, it is still not clear to me why continuous RL is a preferred framework. Finetuning diffusion models in continuous SDEs has been widely studied in the literature. But regarding continuous RL, it is still not made entirely clear why it is needed.
In the Figure 1, it is claimed that "discretizing into 50 steps has more overfitting than discretized with 25 or 100 steps". This is confusing. If I understand correctly, overfitting refers to 'mode collapse'. However, such a claim is not evident. How many runs have you conducted to draw this plot? Is it just for one run? Besides, the reward differences are not significant. Finally, even if this plot is properly drawn, what is the goal of this plot anyway? As far as I am concerned, in practice no matter what continuous formulation is used in modeling, in the final inference stage you would need to choose a discretization number of steps to denoise samples.
In Figure 7, it seems the claim is model fine-tuned by continuous RL and does not exhibit performance difference when using discretization steps. But it is not entirely convincing to me as the experiments can have randomness. Besides, in practice if one discretization step works well it is already sufficient, what is the actual motivation to have several discretization steps with similar performances?

2. The implications and interpretations of Figure 2 are confusing. The authors compare several value network architectures. Firstly the training of value networks with MC across time steps is standard in RL finetuning. The novelty seems to be showing the representative power of introducing sin functions. However I dont think this representativeness can be definitely generalized to all domains of diffusion models. For example if the goal is to train a value function for sequence diffusion model, there are some embedding forms that might outperform sin and cos. Such a randomness and specialty makes me doubt the meaning of showing figure 2 and table 1, which are not generalizable.

**Essential References Not Discussed:**

NA

**Experimental Designs Or Analyses:**

A major concern is about the baseline selection. If the continuous-time RL method proposed in this work intends to demonstrate superiority over existing fine-tuning methods, only comparing against DDPO appears insufficiently justified, as DDPO is relatively slow and computational heavy (Besides, Figure 9 does not seem to indicate a significant difference).
Why not include comparisons against common RL-finetuning baselines such as DRaFT and AlignProp , which are already effective on simple "discrete RL"? Even if the original reward is nondifferentiable, the parametrized value functions would be differentiable so implementation would not be a problem. It is advised to test if continuous RL can further boost a reward direct propagation-based method.

Additionally, the authors completely omit comparisons to a line of inference-time techniques, i.e., Sequential Monte Carlo (SMC)-based methods [1-2]. These inference-time techniques are critical baselines since they do not even incur the computational overhead associated with fine-tuning and can still have good performances. At the cost of fine-tuning, it is necessary to show better performances over the finetuning-free ones.

At minimum, clarifying clear superiority over these other potential baselines would substantially strengthen the importance of using 'continuous RL' versus ordinary "discrete RL".

[1] https://arxiv.org/pdf/2402.06320 [2] https://arxiv.org/pdf/2408.08252

**Methods And Evaluation Criteria:**

The methods and evaluation criteria generally make sense

**Other Comments Or Suggestions:**

1. Eq 10-12 builds that finding the optimal score becomes a policy optimization problem in RL. However, this is known in the literature; see examples [1-2]. References are lacking. Additionally, how are these results different from the known results? Similarly, Thm 3.1 is also a known result.  These take up much space in the main text.
Can the authors comment on why they are novel and what the underlying difficulty of proving them is?

2. It is recommended to enhance interpretations of results (make figures and tables clear in their limitations as much as possible to avoid over-statements that might not definitely be true in other scenarios), add essential experimental baselines (at least explain why some important ones are missing).

[1] https://arxiv.org/pdf/2402.16359 [2] https://arxiv.org/abs/2403.06279

**Other Strengths And Weaknesses:**

Strength:

1. I appreciate the theoretically sound framework of policy optimization in the continuous RL formulation presented in Section 4. The contents are self-contained and implicative for future potential works studying continuous RL settings.

**Questions For Authors:**

see above

**Relation To Broader Scientific Literature:**

NA

**Theoretical Claims:**

Claims are mostly sound.

---

> ### Author Rebuttal · Authors · 2025-03-31
>
> We sincerely thank you for providing detailed and thoughtful feedbacks to improve our paper. Please find our responses to your questions below:
>
>  ***Q1: Clarification on the motivation in Figure 1 and Continuous-RL framework***
>
> ***A1***:  We are sorry if our wording has caused confusion, but we didn't claim on such of "mode collapse". What we claimed  or observed is: the discrete RL trained policy, as a diffusion model, "overfits" to using 50 time steps as inference steps, as illustrated in Figure 1. Diffusion models has a nice natural tradeoff of generation quality and speed, as more inference steps could typically lead to more satisfactory images while causing extra time for inference. The most advanced models typically still haves such tradeoffs. We are motivated to let the fine-tuned models not lose such property (which is a dimension of freedom for users of different purposes), as it's pitiful that the performance improvement brought by RL fine-tuning like DDPO when inferencing the fine-tuned models on other time discretization steps like 25 and 100 could be discounted compared to the improvement typically reported only on fixed time steps like 50.
>
> Comparing our framework to [1] and [2], we agree that the continuous-time control formulation already exists and is similar to our learning objective, but 1) we adopt different parameterizations that focus on treating the score function as the first class citizen and as the primary action space, which motivates us to incorporate the structural property in value function design, 2) the methods in these two works are pure stochastic optimal control methods, which are fundamentally different from our RL-based approaches. The effectiveness and scalability of RL methods have been largely investigated in LLM fine tuning which makes them quite popular, and we believe that our work can help improve the study of RL-based fine-tuning methods for diffusion models by the continuous-time framework with better leverage of structural property and thus resulting scalable, more efficient and more stable algorithm.
>
>  ***Q2: The implications and interpretations of Figure 2***
>
> ***A2***: To summarize, Figure 2 and Table 1 mainly ablate on how to smartly incorporate the diffusion model policy network in the value function design space and partial effects of the link function. As we already theorectically explained our motivations after Equation (20), we conduct experiments in Figure 2 and Table 1 to empirically showcase how things differ when putting the denoised mean $\hat{x}_{\theta}(t,x)$ in the first reward mean predictor term or the second residual corrector term. Our experiments indeed showcase the promise of incorporating such structural property in our design space, as shown by the reduced MSE of green line compared to the orange line. Incoporating the denoised mean in the second part almost does not improve performance while introducing extra computational burden. The performance of the green line and purple line also showcases that the choice of link function also matters. We agree that there could be other link functions that might outperform sin and cos, however it's impossible for us to exhaust all. We would like to pursue possible theorectical guidance of chosing this as future work.
>
> ***Q3: Comparisons against common RL-finetuning baselines such as DRaFT and AlignProp.***
>
> ***A3***: We would like to remark that DRaFT and AlignProp are algorithms for fine-tuning diffusion models, but they are not RL-finetuning methods, which might partically deviate from our primary study on RL-based methods. Nevertheless, we agree that they are baselines for fine-tuning diffusion models and we conduct additional experiments comparing our method, DDPO, DRaFT and AlignProp. Since DRaFT code is not released, we implemented it by revising the codebase of AlignProp. Please find the reward curves comparisons [in the link here](https://ibb.co/KxX6SxhF). Our experimental results show that DRaFT and AlignProp perform similarly (the time costs are also similar), but they both underperform our CTRL approach, or even DDPO when using ImageReward as both the reward signal and the evaluation metric. This demonstrates the advantage of our proposed method.
>
> ***Q4: If continuous RL can further boost a reward direct propagation-based method***
>
> ***A4***: Thank for raising up a direction that could be insightful. Since DRaFT and AlignProp are not RL-based algorithms, we agree and believe that such methods could have continuous time extensions. This is by high level related to our paper of utilizing continuous-time RL over discrete RL, but we believe that the overall framework will be quite different and will not be RL-related anymore. We would add this point to possible direction which can further showcase the insights brought by our framework; we would like to pursue this as future work.

---

> > ### Comment · Reviewer_2RBC · 2025-04-03
> >
> > Thanks for the clarifications. Most of my concerns have been addressed. I have raised the score.
> > I would suggest adding necessary references, interpretations of figures, and comparisons with RL-based finetuning techniques in the next version for completeness and clarity.

---

### Official Review · Reviewer_4zQ6 · 2025-03-12

**Overall Recommendation:** 3

**Summary:**

The paper reformulates fine-tuning diffusion generative models as a continuous-time reinforcement learning problem by treating the score function as a control action in the backward SDE. This continuous-time framework enables direct computation of policy gradients and leads to novel continuous-time analogs of popular RL algorithms (like TRPO/PPO), which are shown to yield more robust and efficient fine-tuning. Additionally, the paper proposes a value network design that leverages the structural properties of diffusion models, and experiments on CIFAR-10 and Stable Diffusion v1.5 demonstrate improved convergence and generation quality over traditional discrete-time methods.

## update after rebuttal
I confirm my score. Authors addressed comments and added clarity and results to the original submission.

**Claims And Evidence:**

In Section 5.2, the paper mentioned that “Continuous-time RL outperforms Discrete-time RL baseline methods in both efficiency and stability because CTRL only require time discretization for estimating the policy gradient. ”, which should provide detaild description and more evidence.

**Essential References Not Discussed:**

No

**Experimental Designs Or Analyses:**

**Section 5.1**

Compare DxMI with CTRL: DxMI is an inverse reinforcement learning based method, which is different from the direct use of reinforcement learning fine-tuning.

**Methods And Evaluation Criteria:**

The methods and evaluation criteria are well-suited to the problem. The continuous-time RL formulation aligns naturally with the intrinsic continuous nature of diffusion processes, and evaluating on benchmarks like CIFAR-10 and Stable Diffusion v1.5 is appropriate for demonstrating improvements in fine-tuning generative models

**Other Comments Or Suggestions:**

Add more experient to show CTRL is practical.

**Other Strengths And Weaknesses:**

**Strengths:**

- A novel continuous time RL finetuning framework

**Weaknesses:**

- Clarity of the paper: add pseudocode, figure of method.
- Small experiments: only compared with 2 models, and add more set of the number of discretization timesteps compared with DPPO.

**Questions For Authors:**

How long does it take of finetune SD v1.5 with CTRL?

**Relation To Broader Scientific Literature:**

The paper related to the field of alignment, diffusion model and reinforment learning.

**Theoretical Claims:**

There are no apparent flaws in the theoretical arguments as presented.

---

> ### Author Rebuttal · Authors · 2025-03-31
>
> We sincerely thank you for providing thoughtful suggestions to improve our paper. Please find our responses to your questions below:
>
>  ***Q1: add pseudocode, figure of method***
>
> ***A1***: Please find the pseucode of our algorithm [here](https://ibb.co/KjhQ4s3m). We will update this part to the revised version.
>
>  ***Q2: provide detaild description and more evidence of "Continuous-time RL outperforms Discrete-time RL baseline methods in both efficiency and stability because CTRL only require time discretization for estimating the policy gradient."***
>
> ***A2***: As illustrated in the earlier Section 4, we are esstentially motivated to approximate the policy gradient formula in Equation (23), since PPO can be understood as more efficient implementations of such policy gradient by introducing the surrogate loss that shares the same gradient and clipping mechanism. This is shown in the definition of loss objective in the [pseucode](https://ibb.co/KjhQ4s3m).
>
> Also we showed in the experiments of fine-tuning stable diffusion v1.5 (the second environment) as our method yields higher rewards training on the same amount of samples compared to DDPO, and the variance of the reward curves also appear to be much smaller. That's why we claim "efficiency and stability", and we will add this heuristic explanation part in the revised version.
>
> ***Q3: only compared with 2 models, and add more set of the number of discretization timesteps compared with DPPO.***
>
> ***A3***: We conduct additional experiments comparing our method, DDPO, DRaFT [1] and AlignProp [2]. Since DRaFT code is not released, we implemented it by revising the codebase of AlignProp. Please find the reward curves comparisons [in the link here](https://ibb.co/KxX6SxhF). Our experimental results show that DRaFT and AlignProp perform similarly, but they both underperform our CTRL approach, or even DDPO when using ImageReward as both the reward signal and the evaluation metric. Due to time constraints, we haven't tested the more set of the number of discretization timesteps of our methods compared with DPPO since they would require more hyparameter tuning for a fair comparison, but we will also add the corresponding results and comparisons in the later version.
>
> ***Q4: How long does it take of finetune SD v1.5 with CTRL?***
>
> ***A4***: The current version of our implemented CTRL takes twice as much time as DDPO for our current implementation, since it requires the value network update and inference which is similar to PPO vs REINFORCE methods in LLMs.
>
> ***References:***
>
> [1] DRaFT: Directly fine-tuning diffusion models on differentiable rewards
>
> [2] AlignProp: Aligning text-to-image diffusion models with reward backpropagation

---

> > ### Comment · Reviewer_4zQ6 · 2025-04-03
> >
> > Thank you for addressing my and other reviewers' comments.

---

### Official Review · Reviewer_b3WA · 2025-03-16

**Overall Recommendation:** 3

**Summary:**

This paper proposes a novel continuous-time reinforcement learning (RL) framework for fine-tuning diffusion models, reframing the learned “score” function as the policy’s action. Unlike discrete-time RL methods for text-to-image (T2I) or other diffusion settings, the paper leverages the underlying continuous-time SDE nature of diffusion models to avoid artifacts from fixed time-discretization and to better exploit structural properties.

**Claims And Evidence:**

Yes, the claims proposed are supported by experiments shown in Section 5.2.

**Essential References Not Discussed:**

N/A

**Experimental Designs Or Analyses:**

The experimental design is generally strong:

- With T=10 steps, they show that continuous-time RL plus carefully chosen sampling steps can drastically improve generation quality (lower FID) compared to a baseline IRL approach.

- Stable Diffusion Fine-Tuning: They measure the ImageReward score across training and across different sampling step counts (25, 50, 100) to confirm time-discretization invariance and consistent improvement brought by the continuous-time RL.

**Methods And Evaluation Criteria:**

Yes. The proposed methods make sense for the problem and the benchmark models are also reasonable (DxMI and SD v1.5).

**Other Comments Or Suggestions:**

N/A

**Other Strengths And Weaknesses:**

Strengths:

- By treating the score as an action in a stochastic control problem, the approach neatly sidesteps time-discretization artifacts.
- The paper presents a clear policy-gradient theorem (Theorem 4.1) and a TRPO/PPO-inspired surrogate bound (Theorem 4.4) specifically for continuous-time, state-independent diffusion coefficients.

Weaknesses:
- The paper only compares its method with DDPO while missing comparisons to other important baselines, such as RAFT and DRaFT, which makes the results less convincing.
- The paper uses only SDv1.5 as the base model; including more models for comparison would improve the evaluation.

**Questions For Authors:**

See above.

**Relation To Broader Scientific Literature:**

The paper belongs to topics related to RL for diffusion models. The paper references existing results on continuous-time RL. The authors refine these ideas for the specific scenario of KL-regularized reward and time-homogeneous diffusion coefficients.

**Theoretical Claims:**

I briefly checked the proofs and they look sound.

---

> ### Author Rebuttal · Authors · 2025-03-31
>
> We sincerely thank you for providing thoughtful suggestions to improve our paper. Please find our responses to your questions below:
>
>  ***Q: Comparisons to other important baselines, such as RAFT and DRaFT***
>
> ***A***: We conduct additional experiments comparing our method, DDPO, DRaFT [1] and AlignProp [2]. Since DRaFT code is not released, we implemented it by revising the codebase of AlignProp. Please find the reward curves comparisons [in the link here](https://ibb.co/KxX6SxhF). Our experimental results show that DRaFT and AlignProp perform similarly, but they underperform our CTRL approach, or even DDPO when using ImageReward as both the reward signal and the evaluation metric.
>
> We would like to comment that we agree these are important diffusion models fine-tuning baseline methods, but our paper primarily focuses on RL-based methods, so we didn't pursue these baseline in our current submitted version. Nevertheless, we agree that these comparisons could strengthen our results, and we will update the experiments of these non-RL baseline methods in the revised version. Due to the time constraints, we haven't tested the RAFT [3] based on ImageReward, but we will also add the corresponding results and comparisons in the later version.
>
> ***References:***
>
> [1] DRaFT: Directly fine-tuning diffusion models on differentiable rewards
>
> [2] AlignProp: Aligning text-to-image diffusion models with reward backpropagation
>
> [3] RAFT:  Reward rAnked Fine-Tuning Algorithm

---

### Decision · Program_Chairs · 2025-05-01

**Decision:**

Accept (poster)

**Comment:**

**Summary:** This work investigates the use of continuous-time reinforcement learning (CTRL) for fine-tuning diffusion models. The authors present a rigorous theoretical framework that establishes connections between CTRL and diffusion fine-tuning, including several insightful lemmas characterizing the KL divergence between state distributions and its relation to the score function. Experimental results demonstrate that CTRL is both robust to discretization strategies and more sample-efficient than existing discrete-time baselines.

**Justification:** All reviewers agree that this work makes a meaningful contribution to the area of diffusion model fine-tuning. However, some reviewers noted confusion in terminology and figure captions. I recommend the authors clarify these points to improve the overall readability and accessibility of the paper.